# Islr regulates canonical Wnt signaling-mediated skeletal muscle regeneration by stabilizing Dishevelled-2 and preventing autophagy

Kuo Zhang[1,2], Yuying Zhang[1], Lijie Gu[1], Miaomiao Lan[1], Chuncheng Liu[1], Meng Wang[1], Yang Su[1], Mengxu Ge[1], Tongtong Wang[1], Yingying Yu[1], Chang Liu[1], Lei Li, Qiuyan Li[1], Yaofeng Zhao [1], Zhengquan Yu [1], Fudi Wang [3], Ning Li[1] & Qingyong Meng [1,2]

Satellite cells are crucial for skeletal muscle regeneration, but the molecular mechanisms regulating satellite cells are not entirely understood. Here, we show that the immunoglobulin superfamily containing leucine-rich repeat (Islr), a newly identified marker for mesenchymal stem cells, stabilizes canonical Wnt signaling and promote skeletal muscle regeneration. Loss of *Islr* delays skeletal muscle regeneration in adult mice. In the absence of Islr, myoblasts fail to develop into mature myotubes due to defective differentiation. Islr interacts with Dishevelled-2 (Dvl2) to activate canonical Wnt signaling, consequently regulating the myogenic factor myogenin (MyoG). Furthermore, Islr stabilizes Dvl2 by reducing the level of LC3-labeled Dvl2 and preventing cells from undergoing autophagy. Together, our findings identify Islr as an important regulator for skeletal muscle regeneration.

[1] State Key Laboratories for Agrobiotechnology, College of Biological Sciences, China Agricultural University, Yuanmingyuan West Road No. 2, Haidian District, Beijing 100193, China. [2] Beijing Advanced Innovation Center for Food Nutrition and Human Health, College of Biological Sciences, China Agricultural University, Yuanmingyuan West Road No. 2, Haidian District, Beijing 100193, China. [3] Beijing Advanced Innovation Center for Food Nutrition and Human Health, College of Food Science & Nutritional Engineering, China Agricultural University, Yuanmingyuan West Road No. 2, Haidian District, Beijing 100193, China. Correspondence and requests for materials should be addressed to Q.M. (email: qingyong.meng@gmail.com)

Regeneration is critical to maintain the homeostasis of adult skeletal muscle in animals, and satellite cells (SCs) play an important role during this process. Adult skeletal muscle fails to regenerate by genetic ablation of SCs postnatally[1–3]. Upon adult skeletal muscle injury, quiescent Pax7[+] SCs are activated and give rise to a population of Pax7[+]/Myf5[+] committed SCs[4,5]. These progenitor cells proliferate into Pax7[+]/MyoD[+] myoblast cells[6,7] and differentiate into the myogenin[+] (MyoG[+]) cells that fuse to form multinucleated myotubes[8,9]. SC differentiation is essential for skeletal muscle regeneration, and it is increasingly clear that SC differentiation is regulated by many signaling pathways. In particular, the canonical Wnt signaling pathway is important for promoting SC differentiation during skeletal muscle regeneration[10,11]. When Wnt ligands bind to Frizzled receptors and members of the low-density lipoprotein receptor-relayed protein (LRP) family, the canonical Wnt signaling is activated. The cytoplasmic part of Frizzled interacts with Disheveled-2 (Dvl2), facilitating interaction between Axin and the LRP tail, which destroys the β-catenin destruction complex and blocks the ubiquitination of β-catenin. Subsequently, β-catenin translocates into the nucleus and further forms a complex with the TCF transcription factors to activate transcription of Wnt target genes[12–14]. Dvl2 is the hub of the canonical Wnt signaling pathway; autophagy mediates the degradation of Dvl2 and further negatively regulates canonical Wnt signaling in response to cellular metabolic stress[15,16]. The deletion of some components of the canonical Wnt signaling pathway delays skeletal muscle regeneration[11]. Inactivation of β-catenin or BCL9 inhibits the differentiation of SCs[17,18]. It is well-known that autophagy is crucial for maintaining the energy balance and stability of the cellular environment[19,20]. Interestingly, autophagy regulates SC activation and skeletal muscle regeneration[21,22]. Given this, we wanted to know whether autophagy regulates muscle regeneration by affecting the stabilization of Dvl2 and how this process is precisely controlled during skeletal muscle regeneration.

Recently, the immunoglobulin superfamily containing leucine-rich repeat (Islr) gene was identified as a new marker of mesenchymal stem cells and is expressed in SCs[23]. Islr mRNA is also detected in muscle tissues[24,25]. It is well-known that Duchenne muscular dystrophy (DMD) patients and dystrophin-null (mdx) mice have continuous regeneration of muscle and activation of SCs in humans and mice, respectively[26–28]. Islr mRNA is highly expressed in DMD patients and mdx mice in the Gene Expression Omnibus (GEO) database, suggesting a potential role of Islr in skeletal muscle regeneration. However, the in vivo functions of Islr in skeletal muscle regeneration are entirely unknown.

In this study, we found that Islr was highly expressed in differentiated myogenic cells. Utilizing an Islr loss-of-function mouse model, we demonstrated that Islr was required for skeletal muscle regeneration. Mechanistically, Islr activated the canonical Wnt signaling pathway by antagonizing autophagy to stabilize Dvl2.

## Results

**Islr is highly expressed in differentiated myogenic cells.** The GEO database shows that Islr mRNA is highly expressed in mdx mice, thus we validated this information and found that Islr was indeed upregulated in mdx mice (Supplementary Fig. 1a, b), indicating that it might be involved in skeletal muscle regeneration. To examine the expression of Islr during skeletal muscle regeneration, the tibial anterior (TA) muscles were injured with an injection of cardiotoxin (CTX) and allowed to regenerate. Islr protein levels were higher in the injured than in the contralateral TA muscles (CTL) at 3 d postinjury (Supplementary Fig. 1c). The

expression level of Islr increased between 3 and 5 day postinjury, which is a critical stage during which SCs participate in skeletal muscle regeneration (Supplementary Fig. 1d). To analyze the expression of Islr during myogenesis, we carried out immunohistochemical staining for Islr, Pax7, and MyoG on serial sections of TA muscles. No Islr protein expression was detected in Pax7[+] quiescent SCs (Fig. 1a). However, Islr protein was expressed in Pax7[+] activated SCs and MyoG[+] muscle progenitors (Fig. 1b, c). A combination of cell surface markers (CD45[−], CD31[−], Sca1[−], and α7-integrin[+]) is widely used to purify SCs in skeletal muscle. Specifically, we isolated SCs from wild-type (WT) mice using fluorescence-activated cell sorting (FACS)[29] (Fig. 1d). Although Islr mRNA was not significantly different between freshly isolated SCs and activated SCs cultured for 24 h, the mRNA and protein levels of Islr were higher in differentiating cells as compared to proliferating cells (Fig. 1e and Supplementary Fig. 1e).

To localize Islr expression in C2C12 cells in vitro, we generated an Islr expression vector in which green fluorescent protein (GFP) was inserted at the C-terminus of Islr (Islr-GFP). Islr co-localized with tubulin following plasmid transfection in C2C12 cells (Fig. 1f), indicating that Islr was expressed in the cytoplasm. Meanwhile, an expression vector fusing a Flag-tag after amino acid 18 (the leader peptide) of Islr was also constructed (L-Flag-Islr), yielding the same result as the Islr-GFP plasmid (Fig. 1g). Meanwhile, we found that endogenous Islr was mainly expressed in the cytoplasm of mature myotubes (Fig. 1h, i and Supplementary Fig. 1f). Furthermore, the expression of Islr gradually increased with the differentiation of C2C12 cells (Fig. 1j, k), which was consistent with the results of primary myoblasts derived from FACS-purified SCs. Taken together, we concluded that Islr was highly expressed in differentiated myogenic cells.

**Loss of Islr delays regeneration of adult skeletal muscle.** In order to study the function of Islr in vivo, we crossed mice possessing the Islr[Exons1b-2-loxP] conditional allele with mice expressing a Cre protein driven in Myf5[+] cells (conditional knockout [cKO])[30] (Fig. 2a). To evaluate the recombination efficiency at the protein level, we purified SCs of control (Ctrl) and Islr cKO mice using FACS and found that Islr was not expressed at all in Islr cKO mice (Fig. 2b). Furthermore, we injected CTX into the TA muscles of control and Islr cKO mice and found that Islr was markedly deleted in most SCs of Islr cKO mice at 5 d postinjury (Fig. 2c, d).

The Islr cKO mice are fertile and healthy without any apparent change in gross phenotype under normal conditions. To understand the role of Islr in skeletal muscle regeneration, we histologically analyzed the TA muscles at 3.5, 5, and 14 d postinjury. At 3.5 d postinjury, control mice already displayed newly formed muscle fibers, whereas almost no muscle fiber regeneration was observed in Islr cKO mice (Fig. 2e and Supplementary Fig. 2a). At 5 d postinjury, control mice had regenerated a large number of muscle fibers with centrally located myonuclei, whereas the diameters of the newly formed muscle fibers were very small in Islr cKO mice (Fig. 2e and Supplementary Fig. 2a). At 14 d postinjury, control mice displayed tightly packed, well-formed muscle fibers, whereas Islr cKO mice had many small muscle fibers with single nuclei (Fig. 2e and Supplementary Fig. 2a). Five days after injury, the average cross-sectional area (CSA) of muscle fibers decreased by about 33% in Islr cKO mice compared to the control mice (Fig. 2f). In addition, the numbers of myofibers containing two or more centrally located nuclei were also significantly reduced in Islr cKO mice (Fig. 2g). We calculated the average muscle fiber areas and found that there were a large number of muscle fibers with an area less than 400 μm² in Islr cKO mice, and muscle

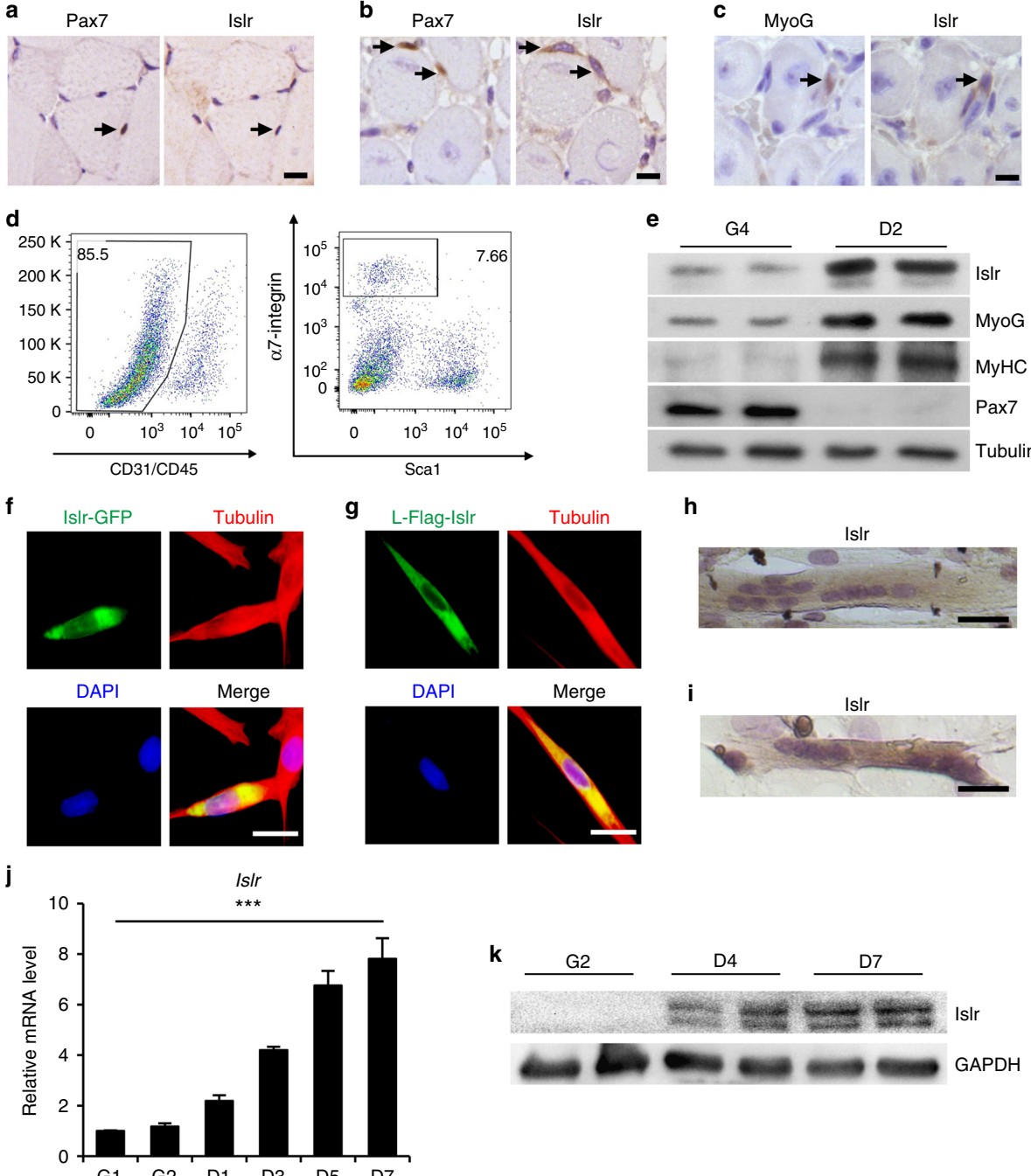

**Fig. 1** Islr is upregulated during satellite cell differentiation and C2C12 differentiation. **a** Immunohistochemistry analysis of Islr and Pax7 in uninjured TA muscles of wild-type (WT) mice. Scale bar = 10 μm. **b** Immunohistochemistry analysis of Islr and Pax7 in injured TA muscles of wild-type mice at 5 d postinjury using serial sections. Scale bar = 10 μm. **c** Immunohistochemistry analysis of Islr and MyoG in injured TA muscles of WT mice at 5 d postinjury using serial sections. Scale bar = 10 μm. **d** Flow cytometry analysis of CD31, CD45, Sca1, and α7-integrin expression in whole muscle-derived cells. **e** Western blot analysis of Islr, Pax7, MyoG, and MyHC in satellite cells from WT mice at 4 d in proliferation medium (G4) or 2 d in differentiation medium (D2). **f** Immunofluorescence staining for tubulin in C2C12 cells transfected with the Islr-GFP plasmid. Scale bar = 25 μm. **g** Immunofluorescence staining for tubulin and Flag in C2C12 cells transfected with the leader peptide-Flag-Islr (L-Flag-Islr) plasmid. Scale bar = 25 μm. **h** Immunohistochemistry analysis of Islr in C2C12 cells cultured in differentiation medium for 5 d. Scale bar = 50 μm. **i** Immunohistochemistry analysis of Islr in primary myoblasts derived from FACS-purified satellite cells cultured in differentiation medium for 2 d. Scale bar = 50 μm. **j** Expression analysis of *Islr* in C2C12 cells cultured in either growth medium for 1 or 2 d (G1 and G2) or in differentiation medium for 1, 3, 5, or 7 d (D1, D3, D5, and D7) using quantitative real-time PCR (qRT-PCR). **k** Western blot analysis of Islr in C2C12 cells cultured in either growth medium for 2 d (G2) or in differentiation medium for 4 or 7 d (D4 and D7). Error bars represent the means ± s.d. ***$P < 0.001$; Student's $t$ test

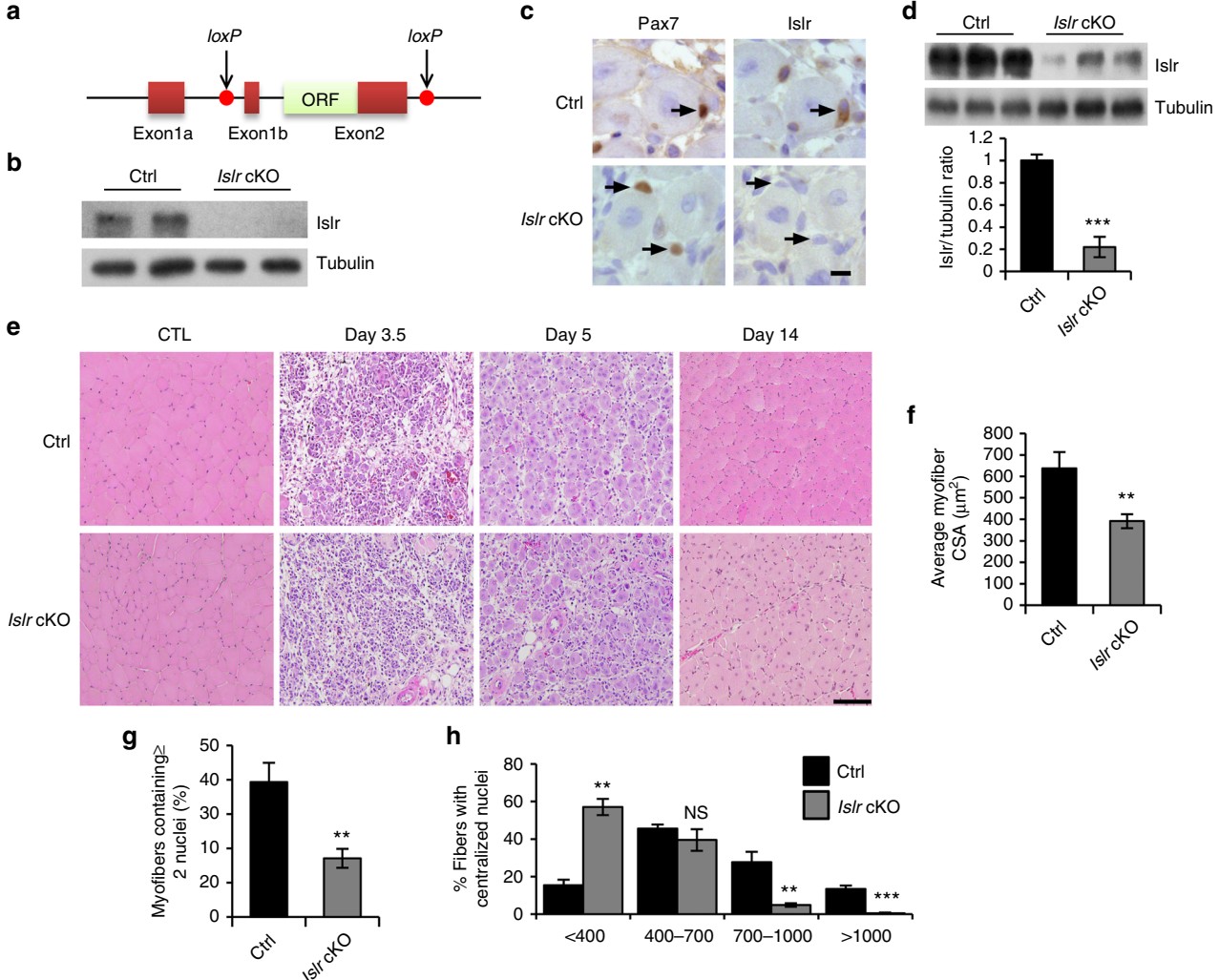

**Fig. 2** The deletion of *Islr* delays skeletal muscle regeneration. **a** A schematic illustration showing the location of *loxP* in the *Islr* gene. **b** Western blot analysis of Islr protein levels in isolated satellite cells of control and *Islr* cKO mice cultured for 4 d in proliferation medium. **c** Immunohistochemistry analysis of Islr in Pax7+ cells of control and *Islr* cKO mice at 5 d postinjury using serial sections. Scale bar = 10 μm. **d** Western blot analysis of Islr protein levels in injured TA muscles of control and *Islr* cKO mice at 5 d postinjury. **e** H&E staining of injured and contralateral TA muscles (CTL) in control and *Islr* cKO mice at 3.5 ($N = 3$), 5 ($N = 4$), and 14 ($N = 3$) d post injury. Scale bar = 100 μm. **f** Average cross-sectional area (CSA) of regenerating myofibers and **g** the number of myofibers containing two or more centrally located nuclei per field at 5 d postinjury. $N = 3$ in each group. **h** Distribution of myofiber CSAs at 5 d postinjury. $N = 3$ in each group. Error bars represent the means ± s.d. NS not significant, $^{**}P < 0.01$, $^{***}P < 0.001$; Student's $t$ test. Control (Ctrl): *Myf5-Cre*$^{+/-}$, *Islr*$^{fl/fl}$; *Islr* cKO: *Myf5-Cre*$^{+/-}$:*Islr*$^{fl/fl}$

fibers with an area greater than 1000 μm² were almost nonexistent in *Islr* cKO mice (Fig. 2h). These results demonstrated that the loss of Islr delayed regeneration of adult skeletal muscle.

**Islr regulates the differentiation potential of SCs**. The higher expression of Islr in differentiated myogenic cells prompted us to investigate the importance of Islr in the process of SCs differentiation during skeletal muscle regeneration. There was a significant reduction in the numbers of eMyHC+ muscle fibers in *Islr* cKO mice compared to control mice at 3.5 d postinjury. At 5 d postinjury, the size of eMyHC+ fibers was significantly reduced in the regenerating TA muscles of *Islr* cKO mice. Statistical analysis showed that the surface area of eMyHC+ muscle fibers was significantly decreased in *Islr* cKO mice (Fig. 3a). This suggested that the differentiation potential of SCs decreased. To further test this idea, we quantified the numbers of MyoG+ cells and found a significant reduction of MyoG+ cells in *Islr* cKO

mice (Fig. 3b). Consistently, Western blotting showed that the protein levels of MyoG and eMyHC were significantly reduced in *Islr* cKO mice at 3.5 and 5 d postinjury (Fig. 3c, d).

To confirm the results in vitro, we isolated single myofibers from the extensor digitorum longus (EDL) muscles of control and *Islr* cKO mice. The number of MyoG+ cells in the clusters associated with single myofibers was greatly reduced in *Islr* cKO mice (Fig. 4a). Next, we analyzed FACS-purified SCs from control and *Islr* cKO mice. Although there was no significant difference during the proliferation period (Supplementary Fig. 2b), we found that the numbers of MyoG+ cells were decreased in *Islr* cKO mice after inducing differentiation (Fig. 4b). Consistently, Western blotting showed that the protein level of MyoG is reduced in *Islr* cKO mice (Fig. 4c). Further, we found that the ability of primary myoblasts derived from FACS-purified SCs to fuse myotubes was significantly decreased, and the fusion index was decreased by about 40% in *Islr* cKO mice compared to controls (Fig. 4d). Statistical analysis showed that *Islr* cKO mice had fewer myotubes containing three or more nuclei (Fig. 4e), and the levels of MyHC

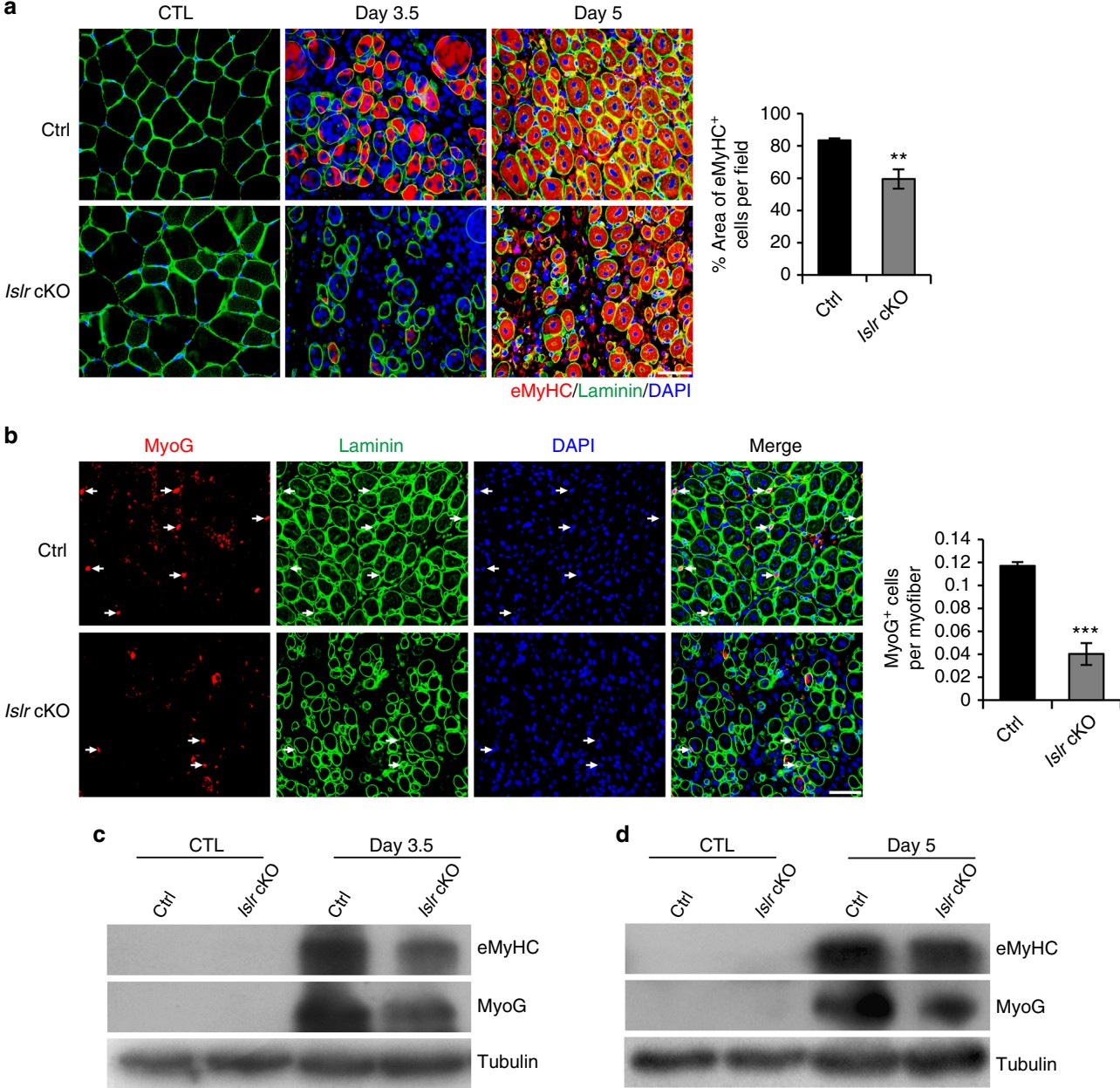

**Fig. 3** Absence of satellite cell differentiation in regenerating TA muscles of *Islr* cKO mice. **a** Immunofluorescence analysis of eMyHC+ fibers in TA muscles of control and *Islr* cKO mice at 3.5 ($N = 3$) and 5 ($N = 3$) d postinjury. The percentage of area occupied by eMyHC+ fibers per field at 5 d postinjury is shown on the right. **b** Immunofluorescence analysis of MyoG+ cells in TA muscles of control and *Islr* cKO mice at 5 d post injury. $N = 3$ in each group. The number of MyoG+ cells per myofiber is shown on the right. **c** Western blot analysis of eMyHC and MyoG protein levels in control and *Islr* cKO mice at 3.5 d postinjury. **d** Western blot analysis of eMyHC and MyoG protein levels in control and *Islr* cKO mice at 5 d postinjury. Scale bars are all 50 μm. Error bars represent the means ± s.d. **$P < 0.01$, ***$P < 0.001$; Student's *t* test

protein were much lower in *Islr* cKO mice (Fig. 4f). To confirm these results, we established primary myoblasts of control and *Islr* cKO mice. Similarly, we observed that the differentiation of primary myoblasts was significantly decreased in *Islr* cKO mice (Supplementary Fig. 2c, d).

**SC-specific deletion of Islr impairs skeletal muscle regeneration.** The *Myf5-Cre* driver functions in the early embryonic development, and may cause compensatory mechanisms to influence the phenotype of *Islr* cKO mice. To address this concern, we crossed mice possessing the *Islr^{fl/fl}* conditional allele with *Pax7-CreER* mice to generate the *Pax7-CreER:Islr^{fl/fl}* mice. The

mice were induced with tamoxifen (TAM) at the age of 10 weeks, followed by CTX damage (Fig. 5a). To evaluate the recombination efficiency at the protein level, we performed immunohistochemistry and Western blot experiments, and found that Islr was markedly deleted in the SCs of *Pax7-CreER:Islr^{fl/fl}* mice at 5 d postinjury (Fig. 5b, c). To test whether *Pax7-CreER:Islr^{fl/fl}* mice had a phenotype consistent with *Islr* cKO mice, we histologically analyzed the TA muscles at 5 and 14 d postinjury. At 5 d postinjury, *Islr^{fl/fl}* mice had regenerated a large number of muscle fibers with centrally located myonuclei, whereas there were many blank areas in the TA muscles of the *Pax7-CreER:Islr^{fl/fl}* mice (Fig. 5d and Supplementary Fig. 2e). Five days after injury, the average CSA of muscle fibers was significantly decreased in *Pax7-*

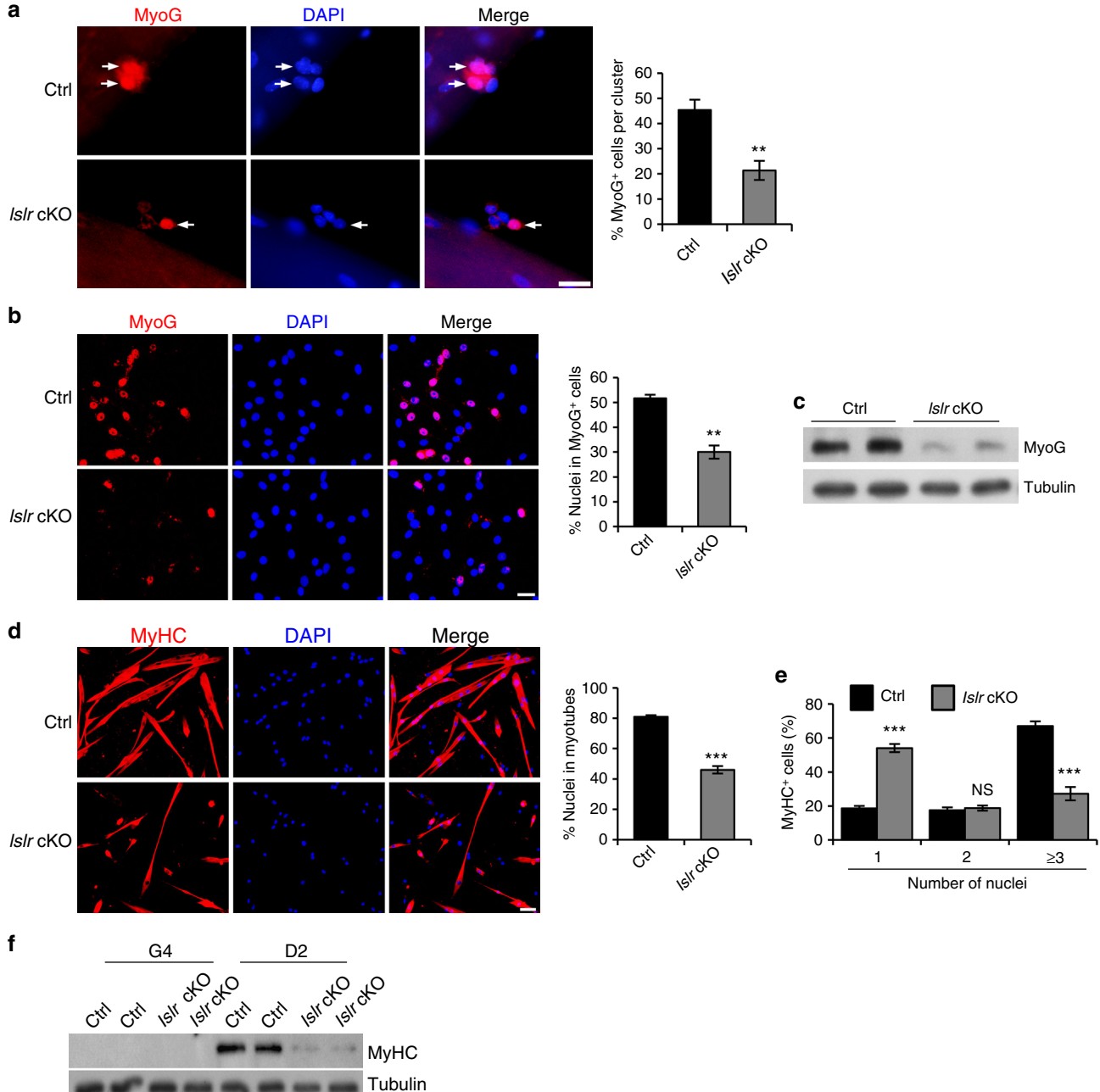

**Fig. 4** The differentiation ability of satellite cells is reduced in *Islr* KO mice in vitro. **a** Immunofluorescence analysis of MyoG$^+$ cells in isolated EDL myofibers of control and *Islr* cKO mice after 3 d of culture. $N = 3$ in each group. Approximately, 30 clusters total in each group. Scale bar = 25 μm. The percentages of MyoG$^+$ cells in the clusters are shown on the right. **b** Immunofluorescence staining for MyoG in FACS-purified satellite cells from control and *Islr* cKO mice cultured for 4 d in proliferation medium followed by 1 d in differentiation medium. $N = 3$ cell cultures in each group. Scale bar = 25 μm. The percentage of MyoG$^+$ cells is shown on the right. **c** Western blot analysis of MyoG protein levels in isolated satellite cells of control and *Islr* cKO mice cultured for 4 d in proliferation medium followed by 1 d in differentiation medium. **d** Immunofluorescence staining for MyHC in FACS-purified satellite cells from control and *Islr* cKO mice cultured for 4 d in proliferation medium followed by 2 d in differentiation medium. $N = 3$ cell cultures in each group. Scale bar = 50 μm. The percentages of nuclei contained in the myotubes (a MyHC$^+$ cell with at least two nuclei) are shown on the right. **e** The distribution of nuclei present in a MyHC$^+$ cell differentiated from FACS-purified satellite cells of control and *Islr* cKO mice. $N = 3$ cell cultures in each group. **f** Western blot analysis of MyHC protein levels in isolated satellite cells of control and *Islr* cKO mice at 4 d in proliferation medium or 2 d in differentiation medium. Error bars represent the means ± s.d. NS: not significant, $^{**}P < 0.01$, $^{***}P < 0.001$; Student's $t$ test

*CreER:Islr$^{fl/fl}$* mice compared to the *Islr$^{fl/fl}$* mice (Fig. 5e). In addition, the numbers of myofibers containing two or more centrally located nuclei were also significantly reduced in *Pax7-CreER:Islr$^{fl/fl}$* mice (Fig. 5f). Meanwhile, the size of eMyHC$^+$ fibers was very small in the regenerating TA muscles of *Pax7-CreER:Islr$^{fl/fl}$* mice (Fig. 5g). At 14 d postinjury, *Islr$^{fl/fl}$* mice

displayed tightly packed, well-formed muscle fibers, whereas *Pax7-CreER:Islr$^{fl/fl}$* mice had many very small muscle fibers with single nuclei (Fig. 5d, h and Supplementary Fig. 2e). Overall, the phenotype of *Pax7-CreER:Islr$^{fl/fl}$* mice was more severe, which demonstrated that Islr had a substantial effect on skeletal muscle regeneration.

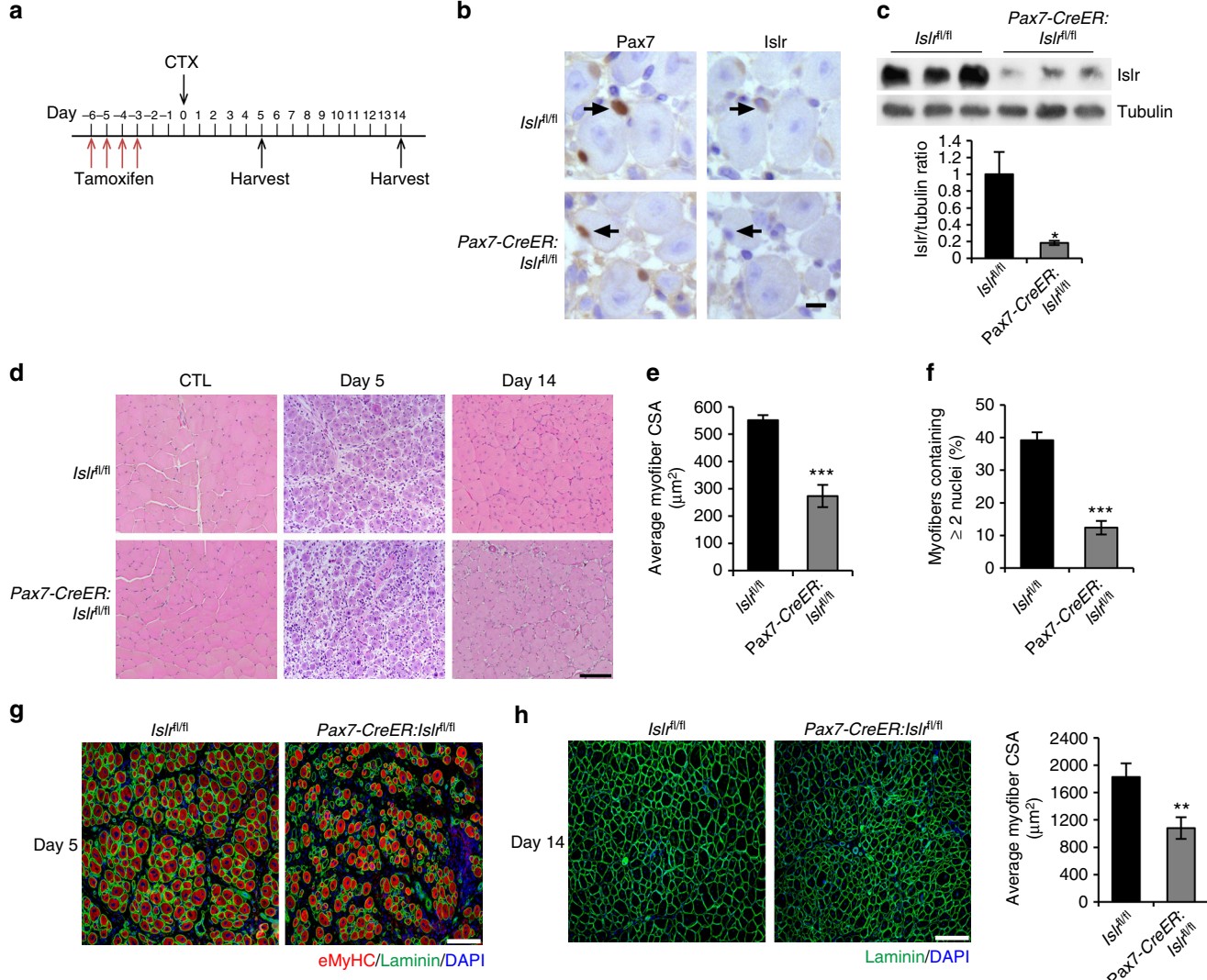

**Fig. 5** Inducible deletion of Islr in satellite cells impairs skeletal muscle regeneration. **a** Schematic outline of CTX injection in tamoxifen-treated *Islr*[fl/fl] and *Pax7-creER:Islr*[fl/fl] mice at the age of 10 weeks. **b** Immunohistochemistry analysis of Islr in Pax7[+] cells of *Islr*[fl/fl] and *Pax7-creER:Islr*[fl/fl] mice at 5 d postinjury using serial sections. Scale bar = 10 μm. **c** Western blot analysis of Islr protein levels in injured TA muscles of *Islr*[fl/fl] and *Pax7-CreER:Islr*[fl/fl] mice at 5 d post injury. **d** H&E staining of injured and CTL in *Islr*[fl/fl] and *Pax7-CreER:Islr*[fl/fl] mice at 5 and 14 d postinjury. N = 3 in each group. Scale bar = 100 μm. **e** CSA of regenerating myofibers and **f** the number of myofibers containing two or more centrally located nuclei per field at 5 d postinjury. N = 3 in each group. **g** Immunofluorescence analysis of eMyHC[+] fibers in TA muscles of *Islr*[fl/fl] and *Pax7-CreER:Islr*[fl/fl] mice at 5 d postinjury. N = 3 in each group. Scale bar = 100 μm. **h** Immunofluorescence analysis of the size of myofibers in TA muscles of *Islr*[fl/fl] and *Pax7-CreER:Islr*[fl/fl] mice at 14 d post injury. N = 3 in each group. Scale bar = 200 μm. The CSAs are shown on the right. Error bars represent the means ± s.d. *P < 0.05, **P < 0.01, ***P < 0.001; Student's *t* test

**Islr functionating by Dvl2-mediated canonical Wnt signaling**. In order to study the molecular mechanisms of Islr during differentiation, we generated *Islr* shRNA stable (sh*Islr*) C2C12 cells. *Islr* mRNA was effectively knocked down in sh*Islr* C2C12 cells compared to control shRNA (shCtrl) C2C12 cells (Supplementary Fig. 3a). The shCtrl and sh*Islr* C2C12 cells were induced to differentiate, and the sh*Islr* C2C12 cells failed to differentiate (Fig. 6a, b). Subsequently, we performed RNA sequencing (RNA-seq) analysis on shCtrl and sh*Islr* C2C12 cells after 2 d of culture in growth medium (G2) or 3 d of culture in differentiation medium (D3). There are 19 signaling pathways whose expression was modulated in both the G2 and D3, in which the Wnt signaling pathway was closely related to C2C12 cells differentiation (Supplementary Fig. 3b, c). Canonical Wnt pathway receptors and downstream target genes were significantly downregulated in sh*Islr* C2C12 cells, and genes related to myogenic development

were also significantly downregulated in sh*Islr* C2C12 cells (Fig. 6c). The downregulation of Wnt target genes and myogenic genes were also confirmed by quantitative real-time PCR (qRT-PCR) (Fig. 6d). These results suggested that the differentiation phenotype of C2C12 cells might be associated with the perturbations of canonical Wnt signaling. Coincidentally, MyoG, a muscle differentiation marker, was reduced in the sh*Islr* C2C12 cells (Fig. 6d). This is significant because MyoG was shown to be directly regulated by the canonical Wnt signaling pathway[31,32].

Consistently, we found that the level of active β-catenin in the nucleus was significantly reduced during the differentiation of sh*Islr* C2C12 cells (Fig. 6e). The TOP/FOP reporter assay also showed that the activity of the canonical Wnt signaling pathway was significantly decreased in the sh*Islr* C2C12 cells (Fig. 6f). Dvl2 is an important component involved in canonical Wnt signaling[16], and Western blotting showed that the protein levels

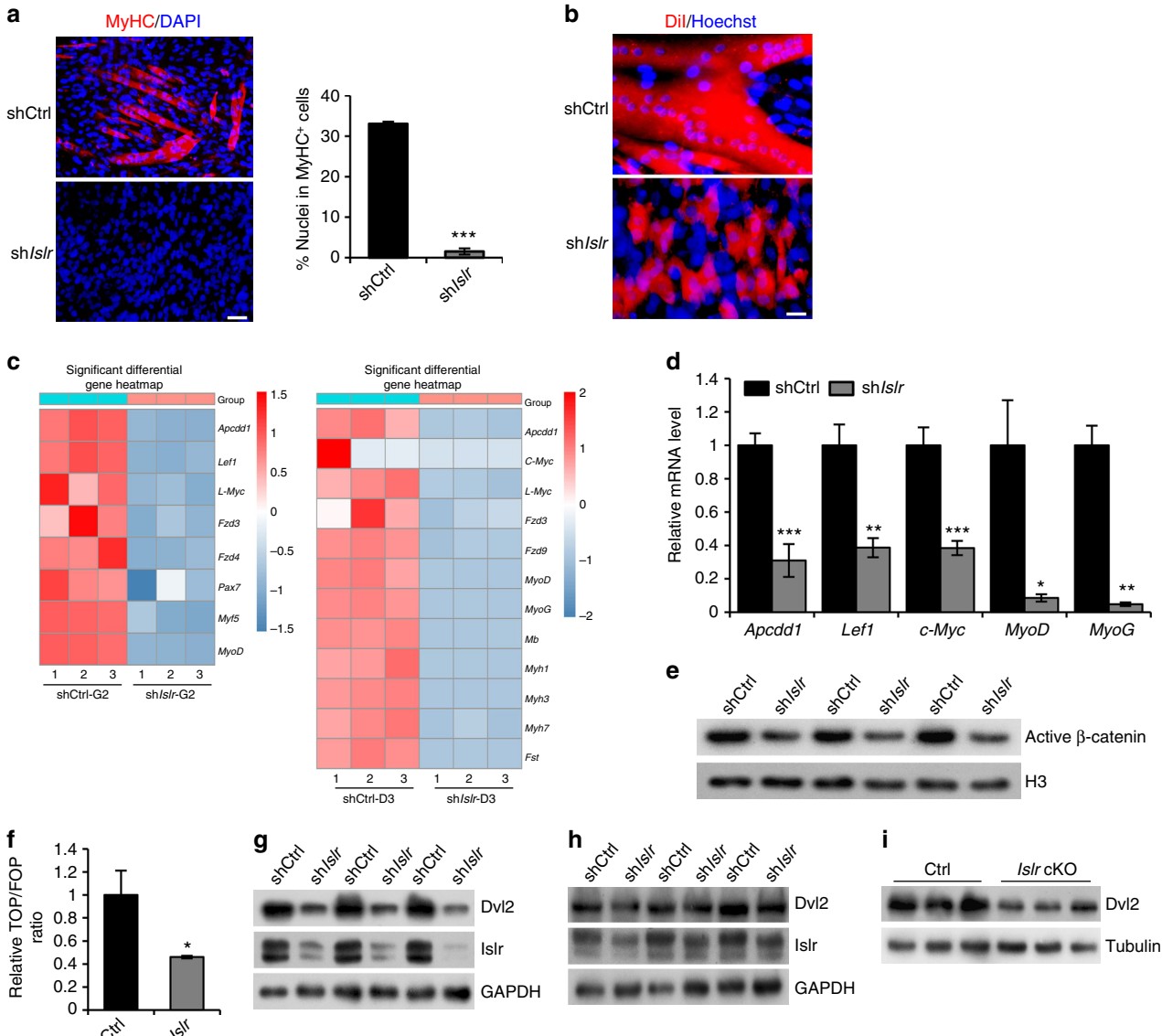

**Fig. 6** The disruption of Islr downregulates the Dvl2-mediated canonical Wnt signaling pathway in myoblasts. **a** Immunofluorescence staining for MyHC in control shRNA (shCtrl) and Islr shRNA stable (shIslr) C2C12 cells. $N = 3$ cell cultures in each group. Scale bar = 50 μm. The percentage of MyHC⁺ cells is shown on the right. **b** Immunofluorescence analysis of cell morphologies in shCtrl and shIslr C2C12 cells by staining live cells with Dil. $N = 3$ cell cultures in each group. Scale bar = 25 μm. **c** Heatmap of the changes in selected gene expression levels in shCtrl and shIslr C2C12 cells at G2 and D3 by RNA-seq. **d** Expression analysis of Wnt target genes and myogenic genes in shCtrl and shIslr C2C12 cells at 3 d in differentiation medium using qRT-PCR. **e** Western blot analysis of active β-catenin protein levels in nuclear lysates extracted from shCtrl and shIslr C2C12 cells after 7 d in differentiation medium. **f** Luciferase activity of TOP/FOP in shCtrl and shIslr C2C12 cells after 3 d in differentiation medium. **g** Western blot analysis of Islr and Dvl2 protein levels in shCtrl and shIslr C2C12 cells after 2 d in growth medium. **h** Western blot analysis of Islr and Dvl2 protein levels in shCtrl and shIslr C2C12 cells after 7 d in differentiation medium. **i** Western blot analysis of Dvl2 protein levels in primary myoblasts of control and Islr cKO mice after 3 d in differentiation medium. Error bars represent the means ± s.d. $^{*}P < 0.05$, $^{**}P < 0.01$, $^{***}P < 0.001$; Student's $t$ test

of Dvl2 were significantly decreased during the proliferation and differentiation of the shIslr C2C12 cells (Fig. 6g, h). Confirming the results obtained with C2C12 cells, we found that the protein levels of Dvl2 were significantly decreased during the differentiation of primary myoblasts from Islr cKO mice (Fig. 6i).

Wnt3a is a classical ligand of the canonical Wnt signaling pathway. Exogenous addition of Wnt3a can activate canonical Wnt signaling. Axin1 was degraded following the addition of Wnt3a in shCtrl C2C12 cells. However, Axin1 still accumulated in shIslr C2C12 cells (Fig. 7a). Meanwhile, the ability of β-catenin to enter the nucleus was significantly reduced in response to Wnt3a addition in shIslr C2C12 cells (Fig. 7b). This suggested

that Wnt3a required Islr for its function. To confirm the results obtained with C2C12 cells, we injected Wnt3a into the TA muscles of control and Islr cKO mice after CTX injury, and found that Wnt3a could not effectively activate canonical Wnt signaling during skeletal muscle regeneration (Fig. 7c, d), which is consistent with the results of C2C12 cells. Furthermore, 1-azakenpaullone (1-AKP), which is known to activate the canonical Wnt signaling pathway independent of Dvl2[33], reduced levels of Axin1 and GSK3β in shIslr C2C12 cells when added to the differentiation medium (Fig. 7e). As a consequence, MyHC was expressed in 1-AKP treated shIslr C2C12 cells (Fig. 7f). To confirm these results, we used another small-molecule Wnt

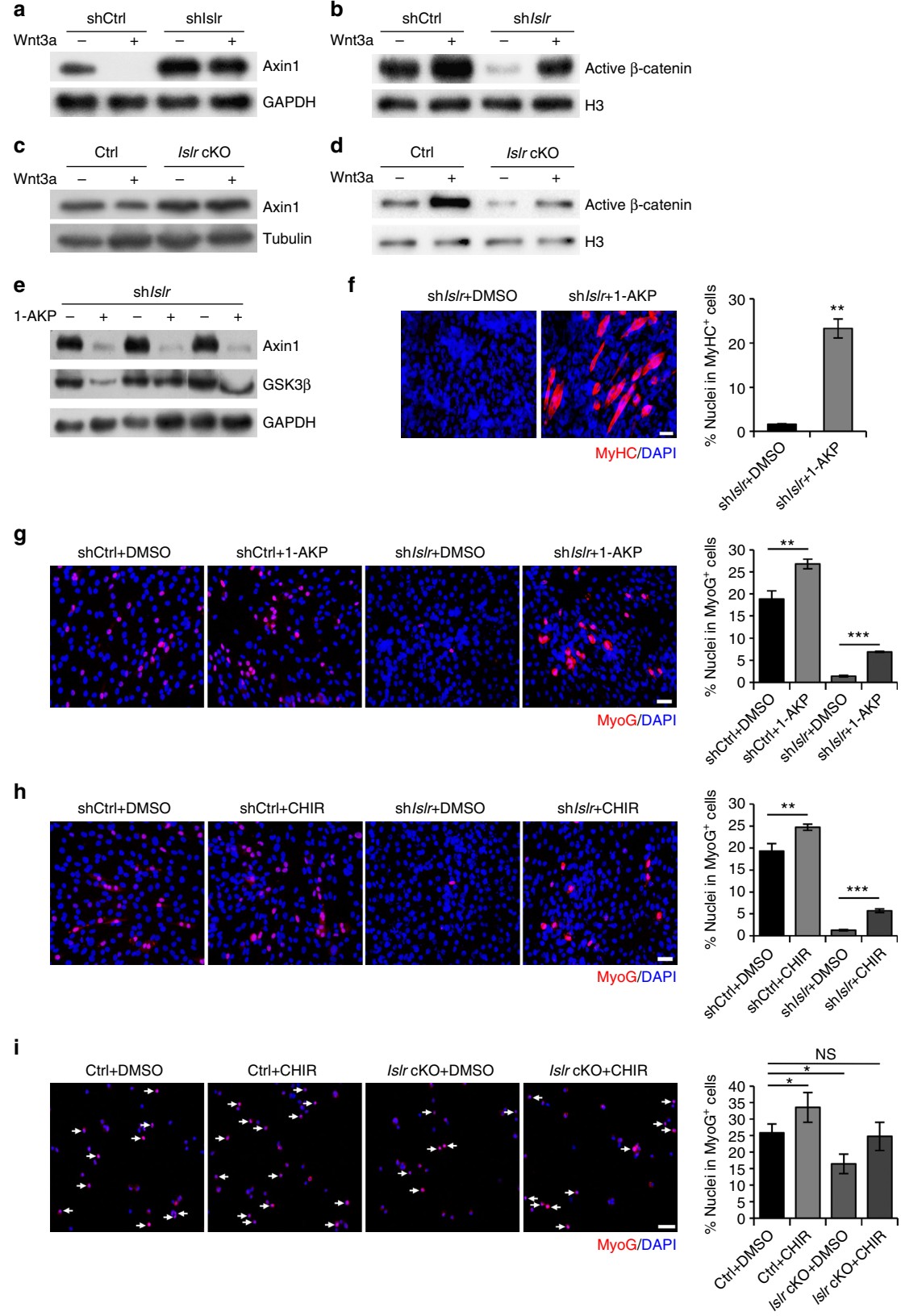

agonist, CHIR-99021 (CHIR)[34], and obtained the same results (Supplementary Fig. 4a). We also checked the expression of MyoG in shCtrl and sh*Islr* C2C12 cells. Not surprisingly, the numbers of MyoG+ cells were significantly decreased in sh*Islr* C2C12 cells (Supplementary Fig. 4b); meanwhile, the protein levels of MyoG were also reduced (Supplementary Fig. 4c). We treated shCtrl and sh*Islr* C2C12 cells with 1-AKP and CHIR, and found that both drugs rescued the MyoG+ cells (Fig. 7g, h). Meanwhile, we obtained the same results from primary myoblasts of control and *Islr* cKO mice that we established (Fig. 7i).

**Fig. 7** The failure of Islr-deficient myoblast differentiation is rescued by canonical Wnt signaling activation. **a** Western blot analysis of Axin1 protein levels in shCtrl and sh*Islr* C2C12 cells incubated with Wnt3a or saline for 24 h after 2 d in differentiation medium. **b** Western blot analysis of active β-catenin protein levels in nuclear lysates extracted from shCtrl and sh*Islr* C2C12 cells incubated with Wnt3a or saline for 24 h after 2 d in differentiation medium. **c** Intramuscular injection of Wnt3a at 1.5 d postinjury and Western blot analysis of Axin1 protein levels in injured TA muscles of control and *Islr* cKO mice at 4 d postinjury. **d** Intramuscular injection of Wnt3a at 1.5 d postinjury and Western blot analysis of active β-catenin protein levels in injured TA muscles of control and *Islr* cKO mice at 4 d postinjury. **e** Western blot analysis of Axin1 and GSK3β protein levels in sh*Islr* C2C12 cells incubated with 1-AKP or DMSO after 7 d in differentiation medium. **f** Immunofluorescence staining for MyHC in sh*Islr* C2C12 cells treated with 1-AKP or DMSO after 7 d in differentiation medium. $N = 3$ cell cultures in each group. The percentages of MyHC$^+$ cells are shown on the right. **g** Immunofluorescence staining for MyoG in shCtrl and sh*Islr* C2C12 cells treated with 1-AKP or DMSO after 3 d in differentiation medium. $N = 3$ cell cultures in each group. The percentages of MyoG$^+$ cells are shown on the right. **h** Immunofluorescence staining for MyoG in shCtrl and sh*Islr* C2C12 cells treated with CHIR or DMSO after 3 d in differentiation medium. $N = 3$ cell cultures in each group. The percentages of MyoG$^+$ cells are shown on the right. **i** Immunofluorescence staining for MyoG in primary myoblasts of control and *Islr* cKO mice treated with CHIR or DMSO after 1 d in differentiation medium. $N = 3$ cell cultures in each group. The percentages of MyoG$^+$ cells are shown on the right. Scale bars are all 50 μm. Error bars represent the means ± s.d. $^*P < 0.05$, $^{**}P < 0.01$, $^{***}P < 0.001$; Student's $t$ test

Cumulatively, 1-AKP and CHIR rescued the phenotype of the sh*Islr* C2C12 cells and the primary myoblasts of *Islr* cKO mice, indicating that Islr might regulate C2C12 cell differentiation via the canonical Wnt signaling pathway.

**Dvl2 complementation rescues the phenotype of *Islr* cKO mice.** The aforementioned data suggested that the loss of Islr blocked the canonical Wnt signaling pathway via Dvl2 (Fig. 6g–i). In order to validate the results obtained from the C2C12 cells, we checked the expression of Dvl2 in TA muscles at 3 d postinjury and found that its expression pattern was consistent with that of Islr (Supplementary Fig. 5a). Meanwhile, Axin1 was also down-regulated, suggesting activation of the canonical Wnt signaling pathway (Supplementary Fig. 5b). Consistently, Dvl2 was highly expressed and β-catenin translocated to nuclei in the cells where Islr was highly expressed at 5 d postinjury (Fig. 8a, b). These results demonstrated that Islr expression correlated with Dvl2. Subsequent results showed that Dvl2 could not be efficiently expressed in *Islr* cKO mice at 3.5 d postinjury (Fig. 8c). Meanwhile, primary myoblasts derived from FACS-purified SCs of *Islr* cKO mice expressed lower levels of Dvl2 and higher levels of Axin1 than control mice after 2 d of culture in differentiation medium. These results suggested that the canonical Wnt signaling pathway was significantly inhibited in *Islr* cKO mice (Fig. 8d). Based on the above results, we injected CHIR into the TA muscles of control and *Islr* cKO mice after CTX injury, and found that CHIR rescued the phenotype of *Islr* cKO mice (Fig. 8e, f). Cumulatively, these data suggested that Dvl2 might be modulated by Islr.

It has been reported that the inhibition of Dact1 upregulates the expression of Dvl2 and activates canonical Wnt signaling[35]. Therefore, we generated *Islr/Dact1* cKO double-mutant mice. Intriguingly, the expression level of Dvl2 was rescued in *Islr/Dact1* cKO mice (Fig. 8g). Moreover, canonical Wnt signaling was reactivated in *Islr/Dact1* cKO mice (Fig. 8h). As a consequence, the deletion of *Dact1* rescued the phenotype of *Islr* cKO mice (Fig. 8i). The CSA of *Islr/Dact1* cKO mice was significantly higher than in *Islr* cKO mice at 5 d postinjury (Fig. 8j). Meanwhile, the number of MyoG$^+$ cells was restored in *Islr* cKO mice (Fig. 8k). Taken together, the in vivo and in vitro data supported the notion that Islr regulated canonical Wnt signaling via modulation of Dvl2.

**Islr stabilizes Dvl2 by antagonizing the autophagy system.** The autophagy system regulates canonical Wnt signaling via modulation of Dvl2 levels[15]. We thus detected the occurrence of autophagy in C2C12 cells using a GFP-LC3 (a well-known marker of autophagosomes) plasmid[36,37]. Overexpression of Dvl2-Flag increased the formation of GFP-LC3$^+$ puncta in

C2C12 cells. However, co-overexpression of Islr-HA and Dvl2-Flag decreased GFP-LC3$^+$ puncta formation compared to overexpression of Dvl2-Flag alone (Fig. 9a). Moreover, co-overexpression of Islr-GFP and Dvl2-Flag decreased the formation of Dvl2-Flag$^+$ puncta compared to overexpression of Dvl2-Flag alone in HEK293T cells (Supplementary Fig. 6a). We found that the autophagy markers LC3II and p62 were significantly increased and decreased, respectively, in sh*Islr* C2C12 cells (Fig. 9b). Furthermore, overexpression of Dvl2-Flag in sh*Islr* C2C12 cells displayed more GFP-LC3$^+$ puncta (Fig. 9c). Moreover, rapamycin (Rap), a well-known autophagy-inducing drug, was used to induce cell autophagy in shCtrl and sh*Islr* C2C12 cells[38,39]. Endogenous Dvl2 formed more GFP-LC3$^+$ puncta in sh*Islr* C2C12 cells (Fig. 9d). To confirm this, we purified SCs from control, *Islr* cKO, *Dact1* cKO, and *Islr/Dact1* cKO mice. After rapamycin treatment, endogenous Dvl2 in primary myoblasts derived from FACS-purified SCs of *Islr* cKO mice formed more GFP-LC3$^+$ puncta, but this was not observed in the *Dact1* cKO mice. Meanwhile, GFP-LC3$^+$ puncta were significantly reduced in *Islr/Dact1* cKO mice (Fig. 9e), which confirmed the previous experiments performed in C2C12 cells. From above results, it was clear that the deletion of *Islr* increased the autophagy-mediated Dvl2 degradation. Therefore, we used bafilomycin A1 (BFA1) to treat primary myoblasts of control and *Islr* cKO mice that we established and found that Dvl2 and Wnt target genes Axin2 and Lef1 were rescued in primary myoblasts of *Islr* cKO mice (Fig. 9f). Meanwhile, the disruption of differentiation was rescued in the primary myoblasts of *Islr* cKO mice when BFA1 was added to the differentiation medium (Supplementary Fig. 6b). To further determine whether Islr interacted with Dvl2 to antagonize the autophagy system, we performed immunoprecipitation analysis and found that Islr-GFP interacted with Dvl2-Flag (Fig. 9g and Supplementary Fig. 6c). Meanwhile, we also found that Islr interacted Dvl2 in differentiated primary myoblasts (Fig. 9h). Dvl2 is ubiquitinated before entering the phagophore[15]. The ubiquitination level of Dvl2 was significantly increased in differentiated primary myoblasts of *Islr* cKO mice, indicating that Islr helped Dvl2 resist ubiquitination (Fig. 9i). Cumulatively, these results indicated that Islr stabilized Dvl2 protein by antagonizing the autophagy system.

## Discussion
Our study found that Islr participated in regulating SC fate and skeletal muscle regeneration. The loss of *Islr* delayed skeletal muscle regeneration in *Myf5-Cre:Islr$^{fl/fl}$* mice. Another SC-specific knockout (*Pax7-CreER:Islr$^{fl/fl}$*) mouse model confirmed these data with an even stronger phenotype. The SC differentiation capacity of *Islr* cKO mice was significantly decreased as a result of the inactivation of canonical Wnt signaling, in which Islr stabilized Dvl2 by antagonizing the autophagy system.

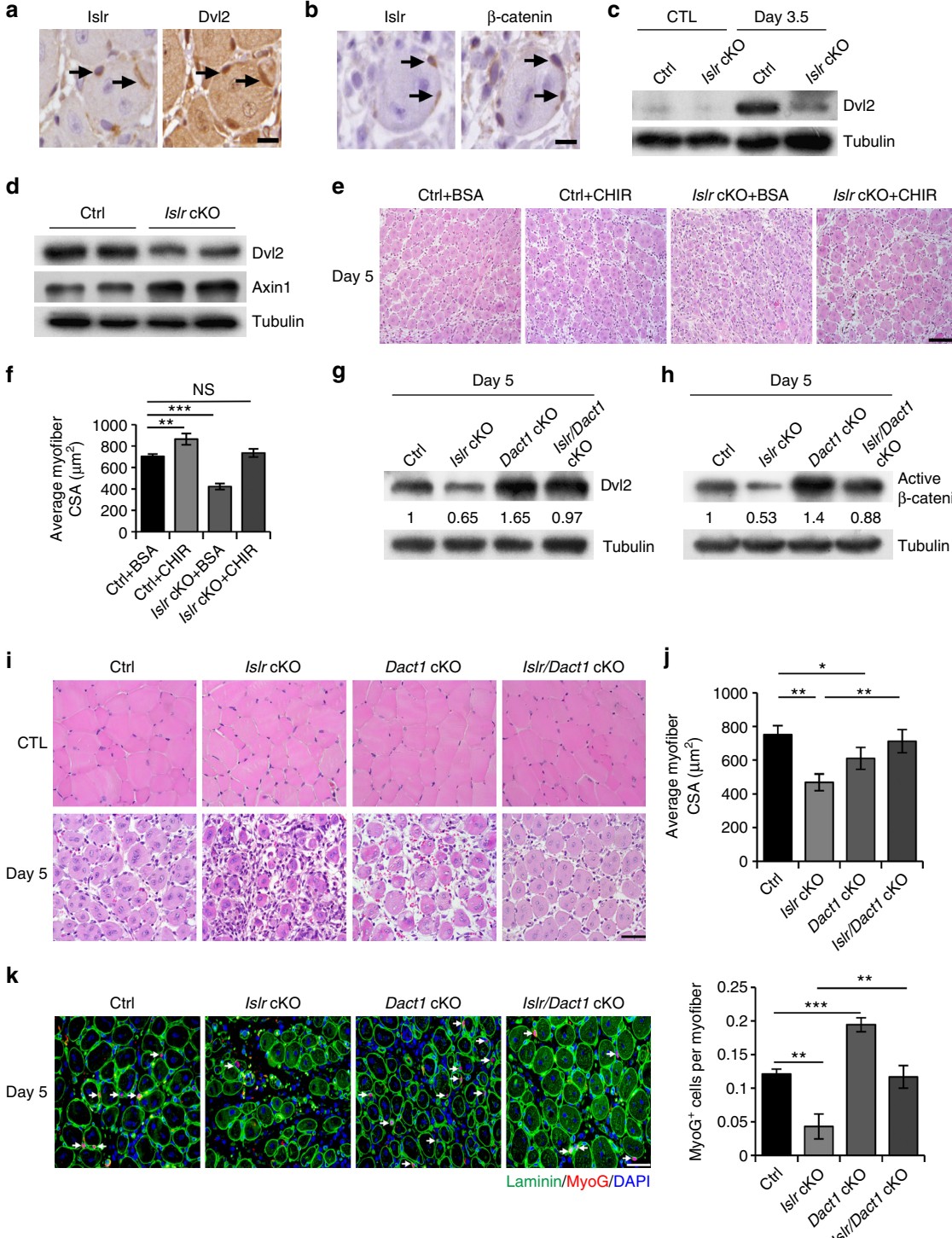

**Fig. 8** The ablation of *Dact1* rescues the impaired skeletal muscle regeneration of *Islr* cKO mice by upregulating Dvl2. **a** Immunohistochemistry analysis of Islr and Dvl2 in injured TA muscles of WT mice at 5 d postinjury using serial sections. Scale bar = 10 μm. **b** Immunohistochemistry analysis of Islr and β-catenin in injured TA muscles of WT mice at 5 d postinjury using serial sections. Scale bar = 10 μm. **c** Western blot analysis of Dvl2 protein levels in injured and CTL of control and *Islr* cKO mice at 3.5 d post injury. **d** Western blot analysis of Dvl2 and Axin1 protein levels in isolated satellite cells of control and *Islr* cKO mice cultured for 2 d in differentiation medium. **e** Intramuscular injection of CHIR or BSA at 2.5 d postinjury and H&E staining of injured TA muscles of control and *Islr* cKO mice at 5 d postinjury. $N = 3$ in each group. Scale bar = 100 μm. **f** CSA of myofibers in (**e**). **g** Western blot analysis of Dvl2 protein levels in injured TA muscles of control, *Islr* cKO, *Dact1* cKO, and *Islr/Dact1* cKO mice at 5 d postinjury. **h** Western blot analysis of active β-catenin protein levels in injured TA muscles of control, *Islr* cKO, *Dact1* cKO, and *Islr/Dact1* cKO mice at 5 d postinjury. **i** H&E staining of injured and contralateral TA muscles in control, *Islr* cKO, *Dact1* cKO, and *Islr/Dact1* cKO mice at 5 d postinjury. Control (Ctrl): $N = 5$; *Islr* cKO: $N = 3$; *Dact1* cKO: $N = 3$; *Islr/Dact1* cKO: $N = 3$. Scale bar = 50 μm. **j** CSA of myofibers in (**i**). **k** Immunofluorescence analysis of MyoG+ cells in TA muscles of control, *Islr* cKO, *Dact1* cKO, and *Islr/Dact1* cKO mice at 5 d postinjury. $N = 3$ in each group. Scale bar = 50 μm. The numbers of MyoG+ cells per myofiber are shown on the right. Error bars represent the means ± s.d. *$P < 0.05$, **$P < 0.01$, ***$P < 0.001$; Student's *t* test

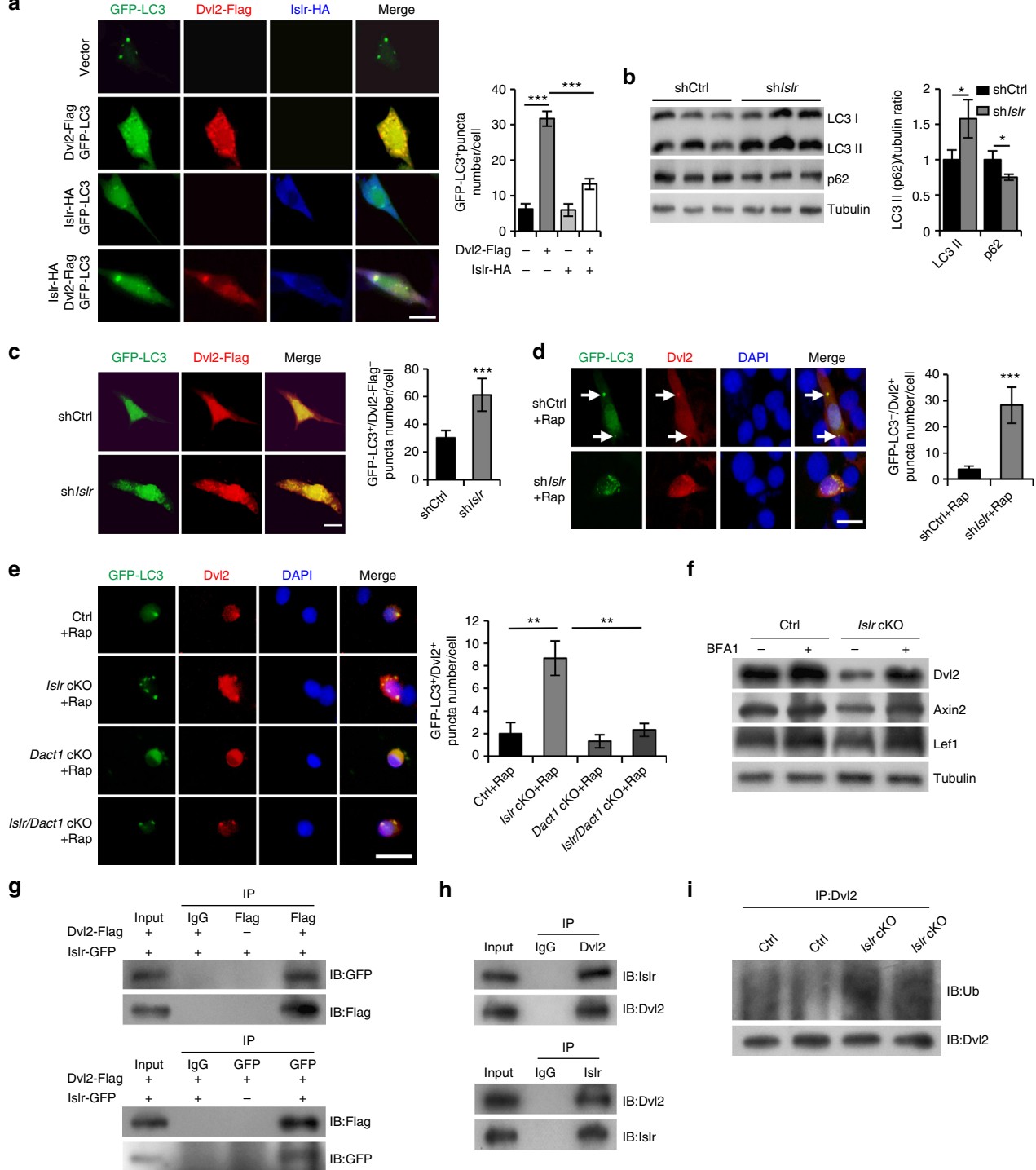

Although we also found that *Islr* mRNA is expressed in quiescent SCs, similar to a recent report[23], we found that Islr protein was not expressed in Pax7+ quiescent SCs. Islr was expressed in Pax7+ activated SCs and more highly expressed in differentiated myogenic cells with the progression of myogenesis, suggesting a potential role for Islr in myogenic differentiation. Supporting this idea, we found that C2C12 cells stopped differentiating into myotubes when disrupting the expression of Islr, and that the ability of myoblasts to differentiate in *Islr* cKO mice was also significantly decreased, as evidenced by the reduction of MyoG in both in vivo and in vitro experiments. It is well-known that MyoG is a key factor required for SC differentiation and

myocyte fusion[40,41]. Skeletal muscle fails to develop normally in the absence of MyoG[8,9]. Taken together, our findings suggest that Islr regulates the differentiation of SCs and C2C12 cells. Although Islr was downregulated during mesenchymal stem cell differentiation into bone and adipose tissues[23], there were many differences in the expressions of genes between these developing tissues and skeletal muscle. It is well-known that bone morphogenetic proteins are expressed in different patterns during bone formation and myogenic differentiation[42–44].

Many reports have shown that the differentiation of C2C12 cells is mainly regulated by canonical Wnt signaling which can directly regulate the expression of MyoD and MyoG[31]. Kindlin 2

**Fig. 9** Islr stabilizes Dvl2 via antagonizing LC3-mediated autophagy. **a** Immunofluorescence analysis of GFP-LC3$^+$ puncta in C2C12 cells transfected with GFP-LC3 and Dvl2-Flag plasmids, GFP-LC3 and Islr-HA plasmids, or GFP-LC3, Dvl2-Flag, and Islr-HA plasmids. $N = 3$ cell cultures in each group. Approximately, 20 total in each group. The numbers of GFP-LC3$^+$ puncta per cell are shown on the right. **b** Western blot analysis of p62 and LC3II protein levels in shCtrl and shIslr C2C12 cells after 2 d in growth medium. **c** Immunofluorescence analysis of GFP-LC3$^+$/Dvl2-Flag$^+$ puncta in shCtrl and shIslr C2C12 cells transfected with GFP-LC3 and Dvl2-Flag plasmids. $N = 3$ cell cultures in each group. Approximately, 20 cells total in each group. The numbers of GFP-LC3$^+$/Dvl2-Flag$^+$ puncta per cell are shown on the right. **d** Immunofluorescence analysis of GFP-LC3$^+$/Dvl2$^+$ puncta in shCtrl and shIslr C2C12 cells treated with rapamycin (Rap) for 6 h. $N = 3$ cell cultures in each group. Approximately, 15 cells total in each group. The numbers of GFP-LC3$^+$/Dvl2$^+$ puncta per cell are shown on the right. **e** Immunofluorescence analysis of GFP-LC3$^+$/Dvl2$^+$ puncta in satellite cells from control, Islr cKO, Dact1 cKO, and Islr/Dact1 cKO mice treated with rapamycin (Rap) for 6 h. $N = 3$ cell cultures in each group. Approximately, 15 cells total in each group. The numbers of GFP-LC3$^+$/Dvl2$^+$ puncta per cell are shown on the right. Scale bars are all 25 μm. **f** Western blot analysis of Dvl2, Axin2, and Lef1 protein levels in primary myoblasts of control and Islr cKO mice incubated with BFA1 or DMSO. **g** Reciprocal co-immunoprecipitation analysis between GFP-tagged Islr and Flag-tagged Dvl2 in HEK293T cells. IB immunoblotting, IP immunoprecipitation. **h** Reciprocal co-immunoprecipitation analysis between Islr and Dvl2 in primary myoblasts cultured for 4 d in differentiation medium. **i** Western blot analysis of the ubiquitination level of Dvl2 in primary myoblasts of control and Islr cKO mice cultured for 4 d in differentiation medium. Error bars represent the means ± s.d. $^*P < 0.05$, $^{**}P < 0.01$, $^{***}P < 0.001$; Student's $t$ test

directly binds the promoter of MyoG in conjunction with β-catenin and TCF4 to regulate C2C12 cell differentiation[32]. Consistent with previous reports, we found that the activity of the canonical Wnt signaling pathway was significantly perturbed in shIslr C2C12 cells, resulting in defective differentiation.

Mechanistically, Islr interacted with and stabilized Dvl2 to activate the canonical Wnt signaling pathway. Dvl2 is an important factor required for canonical Wnt signaling[16,45], and a reduction in Dvl2 expression abrogates canonical Wnt signaling[46]. Confirming this, activation of canonical Wnt signaling independent of Dvl2 was able to rescue the impaired skeletal muscle regeneration phenotype and increase the numbers of MyoG$^+$ cells both in the shIslr C2C12 cells and primary myoblasts of Islr cKO mice. Further, the deletion of Dact1 also rescued the impaired differentiation phenotype of Islr cKO mice by upregulating Dvl2 in vivo. In brief, Islr regulated Dvl2-mediated canonical Wnt signaling and further regulated SC differentiation and skeletal muscle regeneration. Additionally, because Dvl2 has some functions in noncanonical Wnt signaling during skeletal muscle regeneration[11], the potential effect of Islr on other members of the Wnt family, including noncanonical Wnt signaling warrant future investigations.

Autophagy is associated with many important signaling pathways such as Wnt, Shh (Sonic Hedgehog), transforming growth factor β, and fibroblast growth factor (FGF)[47,48]. A key factor is that LC3 targets Dvl2, thus regulating the activity of the canonical Wnt signaling pathway through the autophagy system[15]. In our experiments, we found that Islr stabilized Dvl2 by inhibiting autophagy. GFP-LC3$^+$/Dvl2-Flag$^+$ puncta formed when the Dvl2-Flag plasmid was transfected into C2C12 cells. However, GFP-LC3$^+$/Dvl2-Flag$^+$ puncta could not form and Dvl2-Flag was uniformly dispersed in the cytoplasm when Dvl2-Flag and Islr-HA were co-transfected. These findings suggested that Islr antagonized the autophagy system, consequently protecting Dvl2 against degradation. Furthermore, we demonstrated that endogenous Dvl2 was more easily identified by LC3 both in shIslr C2C12 cells and Islr-deficient primary myoblasts derived from FACS-purified SCs when autophagy was induced.

Previous reports have shown that Dact1 can induce autophagy of Dvl2 and further inhibit canonical Wnt signaling[35,49]. In agreement with this, we found that the deletion of Dact1 inhibited the formation of Dvl2$^+$ autophagic corpuscles in Islr-deficient primary myoblasts derived from FACS-purified SCs. BFA1, an inhibitor of autophagy, rescued Dvl2 protein level in primary myoblasts of Islr cKO mice. Reciprocal co-immunoprecipitation analysis demonstrated that Islr interacted with Dvl2 to antagonize the ubiquitination of Dvl2. It will be interesting to investigate how Islr affects the ubiquitination of Dvl2.

In conclusion, we found that Islr played an important role in the differentiation of SCs during skeletal muscle regeneration. At the molecular level, Islr protected Dvl2 against autophagy-mediated degradation, consequently activating the canonical Wnt signaling pathway (Fig. 10).

## Methods

**Animals**. All experiments were performed in accordance with the China Agricultural University's regulations for animal care and handling. The China Agricultural University Laboratory Animal Welfare and Animal Experimental Ethical Committee authorized the study and approved the protocol. We used the following mouse lines: Myf5-Cre, Pax7-CreER, Islr$^{fl/fl}$, Dact1$^{fl/fl}$. Myf5-Cre and Pax7-CreER mice were provided by Dr. Dahai Zhu (State Key Laboratory of Medical Molecular Biology, Institute of Basic Medical Sciences, Chinese Academy of Medical Sciences, and School of Basic Medicine, Peking Union Medical College) and Dact1$^{fl/fl}$ was provided by Dr. Yeguang Chen (State Key Laboratory of Biomembrane and Membrane Biotechnology, Tsinghua University). Islr$^{fl/fl}$ mice were built by Shanghai Biomodel Organism Science & Technology Development Co., Ltd. All mice were in the C57BL6 background, and their genotype was determined by PCR from tail DNA using the following primers: Myf5-Cre and Pax7-CreER forward: 5′-gcctgcattaccggtcgatgc-3′, Myf5-Cre and Pax7-CreER reverse: 5′-cagggtg ttataagcaatccc-3′; Islr$^{fl/fl}$ forward: 5′-gcgagcaatccagtcctta-3′, Islr$^{fl/fl}$ reverse: 5′-cct gttctgttcaaactatccc-3′; Dact1$^{fl/fl}$ forward: 5′-taccaaggaactacaactcagcc-3′, Dact1$^{fl/fl}$ reverse: 5′-gcccttttagatcaaaatgctactg-3′. To induce Islr deletion in Pax7-CreER:Islr$^{fl/fl}$, tamoxifen (TAM; Sigma) dissolved in corn oil (Sigma) was applied intraperitoneally on four subsequent days to 10-week-old mice at 40 mg/kg.

**Skeletal muscle injury and paraffin sections preparation**. To induce injury, 100 μl of cardiotoxin (CTX, 10 μM in PBS, Sigma) was injected into hindlimb TA muscles. All CTX injured mice were 10 weeks old. The TA muscles were harvested at 3.5, 5, and 14 d postinjury. The TA muscles were fixed with 4% paraformaldehyde for more than 72 h and then subjected to dehydration embedding. Finally, paraffin sections of the muscles were obtained at a thickness of 2–4 μm.

**SC purification and culture**. The hindlimb muscles were minced and digested with 0.2% type II collagenase (Sigma) and 2.5 U/ml dispase (Roche) for 1 h. Each sample was consecutively filtered through 70- and 40-μm cell strainers. The cell suspension was then centrifuged at 600g. Finally, the pellet was washed with FACS buffer (3% FBS in PBS) and stained with CD31-FITC (eBioscience, cat. no. 11-0311-82), CD45-FITC (eBioscience, cat. no. 11-0451-82), Sca1-PerCP (eBioscience, cat. no. 45-5981-82), and α7-integrin-APC (AbLab, cat. no. 67-0010-05) for 1 h at 4°C. We obtained SCs, followed by CD31$^-$, CD45$^-$, Sca1$^-$, and α7-integrin$^+$ cells. FACS-purified SCs were cultured in 24-well plates (NUNC) in Dulbecco's modified Eagle medium (DMEM; Gibco) with 20% FBS (Gibco) and basic FGF (bFGF; 5 ng/ml, Invitrogen). To induce the differentiation, proliferating cells were incubated with DMEM supplemented with 2% horse serum (HS; Gibco).

**Primary myoblast purification and culture**. Primary myoblasts of control and Islr cKO mice were established by the method of differential adhesion. We cultured primary myoblasts in Ham's F10 nutrient mixture medium (Gibco) containing 20% FBS and 5 ng/ml bFGF. To induce differentiation, primary myoblasts were incubated with DMEM supplemented with 2% HS.

**Single myofiber isolation and culture**. The EDL muscles were isolated and digested with 0.2% type I collagenase (Sigma) in DMEM for 1 h. Single myofibers

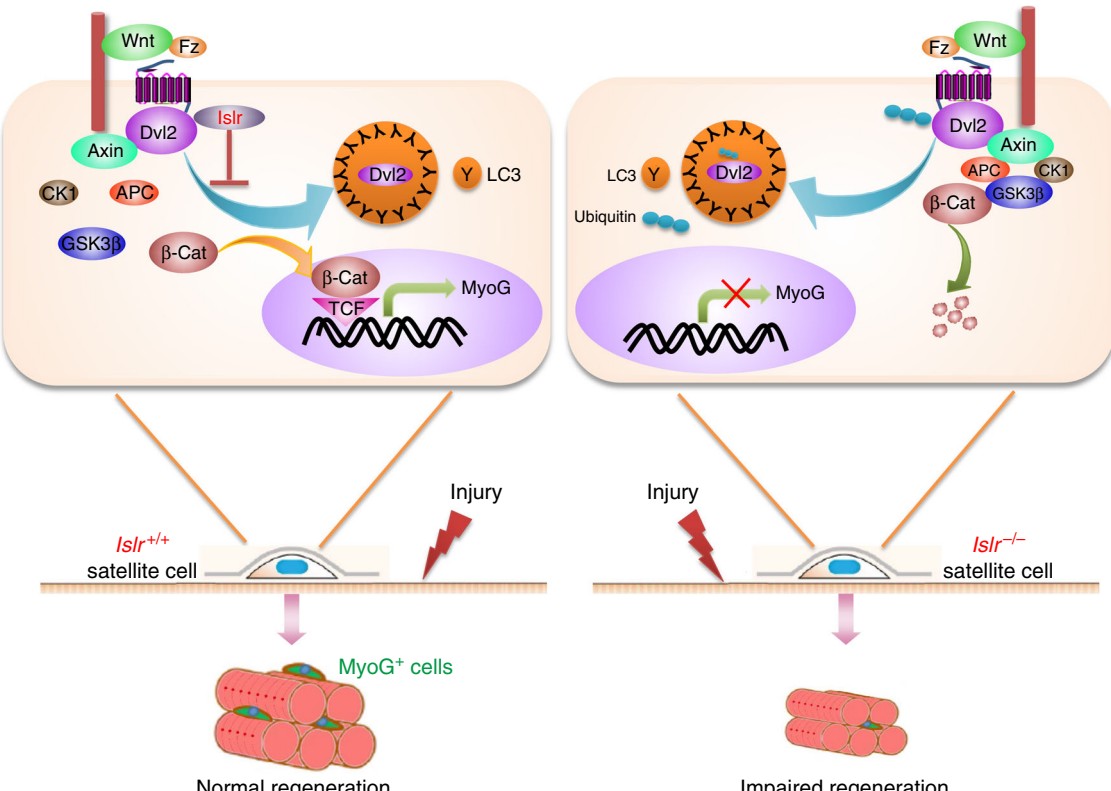

**Fig. 10** Model of the role of Islr during adult skeletal muscle regeneration. Islr protects Dvl2 against autophagy-mediated degradation, resulting in the precise regulation of canonical Wnt signaling during skeletal muscle regeneration. $Islr^{+/+}$: control mice; $Islr^{-/-}$: Islr cKO mice

were transferred to 48-well plates and incubated in DMEM supplemented with 10% HS, 5 ng/ml bFGF, and 0.5% chicken embryo extract (MP Biomedicals).

**Plasmids and RNA interference.** In order to construct a plasmid encoding Dvl2-Flag, the Flag-tag was added at the C-terminus of Dvl2. Dvl2-Flag was then cloned into the BamHI and EcoRI sites of the pcDNA3.1(+) vector (Invitrogen). In order to construct a plasmid encoding Islr-HA, the HA tag was added at the C-terminus of Islr. Islr-HA was then cloned into the BamHI and EcoRI sites of the pcDNA3.1 (+) vector. In order to construct a plasmid encoding leader peptide-Flag-Islr, the Flag-tag was added after amino acid 18 (the leader peptide of Islr). The leader peptide-Flag-Islr was then cloned into the BamHI and EcoRI sites of the pcDNA3.1 (+) vector. The GFP-LC3B plasmid was provided by Dr. Yeguang Chen (State Key Laboratory of Biomembrane and Membrane Biotechnology, Tsinghua University). The TOPflash/FOPflash and *Renilla* luciferase expression plasmids were kept in our lab. Islr-GFP (MG206818), scrambled shRNA, and *Islr* shRNAs (TG512833) were purchased from OriGene. The sequences of *Islr* shRNAs were as follows: *Islr* shRNA1, 5′-cctcagccggtgtctattccagaacaggac-3′; *Islr* shRNA2, 5′-gccaatgtgacca-cactgagcctgtcagc-3′; *Islr* shRNA3, 5′-ggcttcgtgctggcactccactgtgacgt-3′; and *Islr* shRNA4, 5′-ggctgctatacggttgacaacgaggtaca-3′.

**Cell lines and *Islr* shRNA stable (sh*Islr*) C2C12 cells.** C2C12 and HEK293T cell lines were purchased from the Chinese Academy of Medical Sciences and the School of Basic Medicine at Peking Union Medical College. To obtain stably expressing control shRNA (shCtrl) and sh*Islr* C2C12 cells, we transfected C2C12 cells with scrambled shRNA plasmid or *Islr* shRNA3 plasmid via electric trans-fection (Lonza). At 24 h after transfection, the cells were diluted and cultured with puromycin (2 μg/ml, Sigma) for resistance screening for 1 week. The clones were then transferred to a 96-well plate to expand the culture. Finally, transfection was confirmed by PCR, and the primers used were as follows: forward: 5′-gaagagg gcctatttcccat-3′, reverse: 5′-gccttccatctgttgctgcg-3′. The efficacy of *Islr* shRNA interference was determined by quantitative real-time PCR (qRT-PCR) and Western blotting.

**Immunofluorescence and live-cell staining.** Muscle sections, single myofibers, and cultured cells were fixed in 4% paraformaldehyde and permeabilized for 10 min in 0.3% Triton X-100 in PBS. The samples were blocked in blocking buffer (Beyotime Biotechnology) for 2 h at room temperature. Primary antibodies were

incubated in blocking buffer at 4 °C overnight. Subsequently, the samples were washed with PBS and stained with the appropriate fluorescently labeled secondary antibodies (Alexa Fluor 350, 488, or 594) for 1 h at room temperature. After washing with PBS, DAPI (Roche) was used to stain nuclei for 3 min. The primary antibodies used were as follows: mouse anti-Pax7 (Developmental Studies Hybri-doma Bank [DSHB], 1:50), mouse anti-eMyHC (DSHB, 1:50), mouse anti-MyoG (cat. no. ab1835, Abcam, 1:200), mouse anti-MyHC (cat. no. M4276, Sigma, 1:500), rabbit anti-laminin (cat. no. L9393, Sigma, 1:200), rabbit anti-Dvl2 (cat. no. ab22616, Abcam, 1:200), rabbit anti-β-tubulin (cat. no. ab6046, Abcam, 1:500), mouse anti-Flag (cat. no. F1804, Sigma, 1:500), and rabbit anti-HA (cat. no. ab9110, Abcam, 1:200). For live-cell staining, DiI (5 μM, Beyotime Biotechnology) and Hoechst 33342 (2 μg/ml, Beyotime Biotechnology) were applied for 20 min.

**Immunohistochemistry.** Muscle sections were fixed in 4% paraformaldehyde and permeabilized for 10 min in 0.3% Triton X-100 in PBS. The samples were treated with 3% H$_2$O$_2$ for 20 min and blocked in blocking buffer for 2 h at room temperature. Primary antibodies were applied in blocking buffer at 4 °C overnight. Subsequently, the samples were washed with PBS and stained with the appropriate biotin-labeled secondary antibody (goat anti-rabbit or anti-mouse) for 1 h at room temperature. After washing with PBS, the samples were stained with horseradish peroxidase (HRP)-labeled streptavidin for 1 h at room temperature. Finally, DAB (ZSGB-BIO) was used for color development. Samples were then washed with water, and hematoxylin was used to stain the nuclei for 5 min. The primary antibodies used were as follows: rabbit anti-Islr (cat. no. HPA050811, Sigma, 1:300), mouse anti-Pax7 (DSHB, 1:50), mouse anti-MyoG (cat. no. ab1835, Abcam, 1:200), rabbit anti-β-catenin (cat. no. 9562s, Cell Signaling Technology, 1:200), and rabbit anti-Dvl2 (cat. no. PA5-27965, Thermo Fisher, 1:200).

**Plasmid transfection and luciferase reporter assay.** Scrambled shRNA, *Islr* shRNA, Dvl2-Flag, Islr-GFP, Islr-HA, or GFP-LC3B plasmids were transfected into C2C12 cells, HEK293T cells, or SCs using Lipofectamine 2000 (Invitrogen). The samples were then harvested between 24 and 36 h after transfection. In order to identify the activity of canonical Wnt signaling, the TOPflash/FOPflash expression plasmids with the *Renilla* luciferase expression plasmid were transfected when shCtrl and sh*Islr* C2C12 cells were cultured after 2 d in the differentiation medium. After 24 h, the reporter activity was measured using the Dual-luciferase Reporter Assay System (Promega).

**RNA-seq**. The total RNA was extracted from shCtrl and sh*Islr* C2C12 cells using an RNeasy Mini Kit (Qiagen) and on-column DNase digestion (RNase-Free DNase Set, Qiagen) to avoid contamination by genomic DNA. The cDNA library construction, sequencing, and transcriptome data analysis were performed by Gene Denovo Biotechnology Co., Ltd. (Guangzhou, China).

**qRT-PCR analysis**. Total RNA was prepared using TRIzol (Invitrogen). After DNase treatment, 2 μg of RNA was used for reverse-transcription, and qPCR was performed using the LightCycler 480 SYBR Green Master Mix (Roche) in an LC480. Duplicates were performed. The primers used were as follows: *Gapdh* (5′-gtgccgcctggagaaacct-3′ and 5′-aagtcgcaggagacaacc-3′), *Islr* (5′-agatccgctcggtggct att-3′ and 5′-aggtcgctccaggcaaact-3′), *Apcdd1* (5′-ctgaagcatctccacaacgg-3′ and 5′-ggacccgaccttacttcaca-3′), *c-Myc* (5′-tagtgctgcatgaggagaca -3′ and 5′-ctccacaga caccacatcaa-3′), *Lef1* (5′-gacagatcaccccacccatt-3′ and 5′-atagctggatgagggatgcc-3′), *MyoD* (5′-gcagaatggctacgacaccg-3′ and 5′-cactatgctggacaggcagt-3′), and *MyoG* (5′-caatgcactggagttcggtc-3′ and 5′-gctgtccacgatggacgtaa-3′).

**Western blot analysis and immunoprecipitation analysis**. C2C12 cells, SCs, and TA muscles were washed with PBS and lysed in RIPA lysis buffer (Beyotime Biotechnology). Next, 200 μg of total protein was resolved by 10% sodium dodecyl sulfate polyacrylamide gel electrophoresis and electrotransferred onto a poly-vinylidene fluoride (PVDF) (Millipore) membrane. The PVDF membrane was blocked in 5% skim milk powder dissolved in TBST for 2 h at room temperature. Primary antibodies were applied in sealing fluid at 4 °C overnight. Subsequently, the PVDF membrane was washed with TBST and stained with the appropriate HRP-labeled secondary antibodies (goat anti rabbit or mouse) for 1 h at room temperature. After washing with TBST, the ECL Reagent (Millipore) was used, and the strips were on film. For immunoprecipitation analysis, the HEK293T cells were lysed with IP lysis buffer (Beyotime Biotechnology), and the total protein was immunoprecipitated with Flag or GFP antibody and protein G beads (Thermo Fisher). Immunocomplexes were washed three times with IP lysis buffer and analyzed by Western blot. The primary antibodies used were as follows: rabbit anti-Islr (cat. no. HPA050811, Sigma, 1:1000), rabbit anti-Axin1 (cat. no. 2087s, Cell Signaling Technology, 1:1500), rabbit anti-GSK3β (cat. no. ab32391, Abcam, 1:1500), mouse anti-MyoG (cat. no. ab1835, Abcam, 1:1000), mouse anti-MyHC (cat. no. M4276, Sigma, 1:2000), rabbit anti-Active β-catenin (cat. no. 8814s, Cell Signaling Technology, 1:1000), rabbit anti-Dvl2 (cat. no. ab22616, Abcam, 1:1000), rabbit anti-Ubiquitin (cat. no. ab134953, Abcam, 1:1500), rabbit anti-VDAC (cat. no. 4866s, Cell Signaling Technology, 1:2000), rabbit anti-LC3B (cat. no. ab51520, Abcam, 1:1500), rabbit anti-p62 (cat. no. ab109012, Abcam, 1:1000), rabbit anti-Axin2 (cat. no. ab109307, Abcam, 1:1500), rabbit anti-Lef1 (cat. no. 2286 s, Cell Signaling Technology, 1:1500), rabbit anti-GAPDH (cat. no. 2118s, Cell Signaling Technology, 1:5000), rabbit anti-β-tubulin (cat. no. ab6046, Abcam, 1:5000), rabbit anti-Histone H3 (cat. no. ab1791, Abcam, 1:5000), mouse anti-Flag (cat. no. F1804, Sigma, 1:2000), and rabbit anti-GFP (cat. no. ab290, Abcam, 1:1500). Uncropped blots for Western blot analysis are shown in Supplementary Fig. 7.

**Autophagy analysis**. C2C12 cells or SCs were transfected with GFP-LC3B plasmid. After 24 h, cells were treated with rapamycin (4 μM, Sigma) for 6 h to induce autophagy, followed by immunofluorescence analysis.

**Treatment with reagents in cell culture**. In order to verify the activity of canonical Wnt signaling, shCtrl and sh*Islr* C2C12 cells were treated with either Wnt3a (100 ng/ml, Peprotech) or saline for 24 h after 2 d in differentiation medium. The cell samples were then collected for Western blot analysis. In order to activate canonical Wnt signaling in sh*Islr* C2C12 cells, the sh*Islr* C2C12 cells were treated with either 1-azakenpaullone (1-AKP; 3 μM, Sigma) or CHIR-99021 (CHIR; 3 μM, Stemgent) during differentiation. Western blotting and immuno-fluorescence analysis were performed after 3 or 7 d of differentiation. To verify the activity of canonical Wnt signaling in vivo, the TA muscles of control and *Islr* cKO mice were injected with Wnt3a (100 ng/ml, Peprotech) or PBS (0.1% BSA) at 1.5 d postinjury. The injured TA muscles were then collected for Western blot analysis at 4 d postinjury. To activate canonical Wnt signaling in vivo, the TA muscles of control and *Islr* cKO mice were injected with CHIR-99021 (50 ng/ml, Stemgent) or PBS (0.1% BSA) at 2.5 d postinjury. The injured TA muscles were then collected for H&E analysis at 5 d postinjury. To inhibit autophagy, the primary myoblasts of control and *Islr* cKO mice that we established were treated with bafilomycin A1 (BFA1; 200 nM, Sigma) for 6 h. Then cell samples were collected for Western blot analysis.

**Statistical analysis**. All experiments included at least three biological replicates. The areas of fluorescent staining for muscle fibers and nuclei were determined using ImageJ software. Data were analyzed using Microsoft Excel. A two-tailed Student's *t* test was used to assess statistical significance.

## Data availability

RNA-seq data used in this study have been deposited in the National Center for Biotechnology Information Sequence Red Archive (SRA) under the accession code SRP160431. All other data are available within the Article and its Supplementary Files, or available from the authors upon request.

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

## Acknowledgments

We thank Dr. Yeguang Chen for the *Dact1fl/fl* mice. We thank Dr. Dahai Zhu and Dr. Yong Zhang for the *Myf5-Cre* and *Pax7-CreER* mice. We thank Dr. Yeguang Chen for the GFP-LC3B plasmid. We thank Dr. Xiangdong Li for revising the manuscript. This work was supported by the National Basic Research Program of China (2015CB943103), the National Natural Science Foundation of China (31790412), the earmarked fund for Modern Agro-Industry Technology Research Systems of China (CARS-36), and the Ministry of Agriculture Transgenic Major Projects of China (2016ZX08010004).

## Author contributions

K.Z. designed and performed the experiments, analyzed the data, and wrote the manuscript. Y.Z., L.G., and M.L. carried out vector construction and mouse rearing. C.L., M.W., Y.S., M.G., T.W., Y.Y., C.L., and L.L. provided technical assistance. Q.L., Y.Z., Z.Y., F.W., and N.L. revised the manuscript. Q.M. guided the experiments, analyzed the results, revised the manuscript, and provided financial support. All the authors edited and approved the manuscript.

## Additional information

**Competing interests:** The authors declare no competing interests.

