## [Peer Review File · Nature Communications]

This manuscript has been previously reviewed at another journal that is not operating a transparent peer review scheme. This document only contains reviewer comments and rebuttal letters for versions considered at Nature Communications. Mentions of the other journal have been redacted

Reviewers' Comments:

Reviewer #1:

Remarks to the Author:

The authors addressed most of the concerns raised by this Reviewer thus improving the quality of the paper.

Nevertheless, the mild phenotype observed in the "muscle-specific" *Islr* ko mice is still mild and could also be due to the contribution derived by surrounding tissues. The experiments developed by the authors in response to my concern about this issue is not exhaustive at all since it has been driven on wt myoblasts (apparently, since no details have been provided).

Reviewer #2:

Remarks to the Author:

The authors have satisfactorily addressed most of the comments by adding new data or discussion. Yet, there are two original questions that remained unanswered. As they are key for the conclusion of the study, these two specific questions need to be addressed, as follows:

1. Ok. Although this indicates a lower impact of *Islr* on Wnt target gene expression and muscle regeneration.
2. Ok.
3. Ok.
4. Not satisfied. As raised in the original comments, to demonstrate the physiological role of autophagy in *Islr* KO/KD cells, the authors should show whether ablation of an autophagy gene(s) can rescue the abnormality in Wnt gene expression and/or regeneration potential in *Islr* null cells. Alternatively, studying the effects of BFA1 treatment on Wnt gene expression and regeneration in *Islr* null cells or animals is also acceptable.
5. Ok. Please include quantification in Fig. 9b.
6. Ok.
7. Ok.
8. Not satisfied. As mentioned in the original comments, the image for "*Islr*-HA + GFP-LC3" is still missing in new Fig. 9a, although it is present in the quantification (last lane). Also, what is the number of GFP-LC3 puncta under the condition of "Dvl2-Flag + GFP-LC3"? There is apparently GFP signal in the image (row 2) but the condition is absent from the quantification.
9. Ok.

Reviewer #3:

Remarks to the Author:

After the second revision of the manuscript, Zhang K. and collaborators have positively improved the robustness of their hypothesis as it was requested. All my questions and specific technical concerns were properly addressed. I want to highlight that all the in vitro data previously performed on C2C12 and HEK293T has been demonstrated on primary myoblast for the second submission, making their manuscript more robust. Also the fact of studying the role of *Islr* with a Pax7 promoter, has noteworthy reinforced the hypothesis of *Islr* as a regulator of muscle

regeneration.

In addition to improving all the technical part, they have improved the abstract, introduction and discussion by adding new literature and a deeper analysis of their data.

Responses to the reviewers' comments:

Reviewer #1:

Comment: The authors addressed most of the concerns raised by this Reviewer thus improving the quality of the paper.

Nevertheless, the mild phenotype observed in the "muscle-specific" Islr ko mice is still mild and could also be due to the contribution derived by surrounding tissues. The experiments developed by the authors in response to my concern about this issue is not exhaustive at all since it has been driven on wt myoblasts (apparently, since no details have been provided).

Response: Thank you very much for your enthusiasm and suggestions for our revised manuscript. We validated your suggestions in all our mouse models. To confirm the experiments of wild type (WT) myoblasts, we established primary myoblasts of control (Ctrl) and *Myf5-Cre:Islr^{fl/fl}* (*Islr* cKO) mice. The disruption of differentiation was not rescued in the primary myoblasts of *Islr* cKO mice when Islr protein was added to the differentiation medium (See the below figure, **a**). Meanwhile, we injected Islr protein into the cardiotoxin (CTX)-injured tibial anterior (TA) muscles of *Islr^{fl/fl}* and *Pax7-CreER:Islr^{fl/fl}* mice at 2.5 d post injury, and found that exogenous Islr protein did not rescue the phenotype of *Pax7-CreER:Islr^{fl/fl}* mice (See the below figure, **b**). Combined with all the experiments using myoblasts, and *Myf5-Cre:Islr^{fl/fl}* and *Pax7-CreER:Islr^{fl/fl}* mice, we conclude Islr has a specific role in satellite cells and muscle regeneration.

a Immunofluorescence staining for MyHC in primary myoblasts of control and *Islr* cKO mice treated with Islr protein or BSA after 3 d in differentiation medium. $N = 3$ cell cultures in each group. Scale bar = 100 μm . The percentages of nuclei contained in the myotubes (a MyHC⁺ cell with at least 2 nuclei) are shown on the right. **b** Intramuscular injection of Islr protein or BSA at 2.5 d post injury and H&E staining of injured TA muscles of *Islr^{fl/fl}* and *Pax7-CreER: Islr^{fl/fl}* mice at 5 d post injury. $N = 3$ in each group. Scale bar = 100 μm . The CSAs are shown on the right. Error bars represent the means \pm s.d. NS: not significant, ** $P < 0.01$; Student's t test. Control (Ctrl): *Myf5-Cre^{+/-}*, *Islr^{fl/fl}*; *Islr* cKO: *Myf5-Cre^{+/-}:Islr^{fl/fl}*.

Reviewer #2:

Comment: The authors have satisfactorily addressed most of the comments by adding new data or discussion. Yet, there are two original questions that remained unanswered. As they are key for the conclusion of the study, these two specific questions need to be addressed.

Response: Thank you very much for your enthusiasm and suggestions for our revised manuscript. We have done our best to address all of your concerns. Please find our detailed responses below.

4. Comment: As raised in the original comments, to demonstrate the physiological role of autophagy in *Islr* KO/KD cells, the authors should show whether ablation of an autophagy gene(s) can rescue the abnormality in Wnt gene expression and/or regeneration potential in *Islr* null cells. Alternatively, studying the effects of BFA1 treatment on Wnt gene expression and regeneration in *Islr* null cells or animals is also acceptable.

Response: Thank you for your suggestion. We established primary myoblasts of control and *Islr* cKO mice. The primary myoblasts of *Islr* cKO mice upregulated the Wnt target genes *Axin2* and *Lef1* when treated with bafilomycin A1 (BFA1) (Fig. 9f). The disruption of differentiation was rescued in the primary myoblasts of *Islr* cKO mice when BFA1 was added to the differentiation medium (Supplementary Fig. 6b).

Supplementary Fig. 6b Immunofluorescence staining for MyHC in primary myoblasts of control and *Islr* cKO mice treated with BFA1 or DMSO for 24 h after 3

d in differentiation medium. $N = 3$ cell cultures in each group. Scale bar = 100 μm . The percentages of nuclei contained in the myotubes (a MyHC⁺ cell with at least 2 nuclei) are shown on the right. Error bars represent the means \pm s.d. * $P < 0.05$, *** $P < 0.001$; Student's t test. Control (Ctrl): *Myf5-Cre*^{+/-}, *Islr*^{fl/fl}; *Islr* cKO: *Myf5-Cre*^{+/-}:*Islr*^{fl/fl}.

5. Comment: Ok. Please include quantification in Fig. 9b.

Response: Thank you for your suggestion. We show the quantification in the new Fig. 9b.

8. Comment: As mentioned in the original comments, the image for “Islr-HA + GFP-LC3” is still missing in new Fig. 9a, although it is present in the quantification (last lane). Also, what is the number of GFP-LC3 puncta under the condition of “Dvl2-Flag + GFP-LC3”? There is apparently GFP signal in the image (row 2) but the condition is absent from the quantification.

Response: We are very sorry that the abscissa axis was mistaken in the quantification of Fig. 9a in the old version. In the old version, lane 1 was GFP-LC3, lane 2 was Dvl2-Flag + GFP-LC3, and the last lane was Dvl2-Flag + Islr-HA + GFP-LC3. We have corrected the error of the abscissa axis of the old Fig. 9a. Meanwhile, we have added an image for Islr-HA + GFP-LC3 and the quantification to the new Fig. 9a in accordance with your comments.

Reviewer #3:

Comment: After the second revision of the manuscript, Zhang K. and collaborators have positively improved the robustness of their hypothesis as it was requested. All my questions and specific technical concerns were properly addressed. I want to highlight that all the in vitro data previously performed on C2C12 and HEK293T has been demonstrated on primary myoblast for the second submission, making their manuscript more robust. Also the fact of studying the role of Islr with a Pax7 promoter, has noteworthy reinforced the hypothesis of Islr as a regulator of muscle regeneration.

In addition to improving all the technical part, they have improved the abstract, introduction and discussion by adding new literature and a deeper analysis of their data.

Response: Thank you very much for your satisfaction with our revised manuscript.

Reviewers' Comments:

Reviewer #1:

Remarks to the Author:

The authors addressed now all the last concerns raised by this Reviewer thus improving the quality of the paper.

Reviewer #2:

Remarks to the Author:

The authors have satisfactorily addressed all the comments. The study tested a novel hypothesis supported by solid data and a clear rationale.

Responses to the reviewers' comments:

Reviewer #1:

Comment: The authors addressed now all the last concerns raised by this Reviewer thus improving the quality of the paper.

Response: We thank Reviewer #1 for his/her enthusiasm and insightful comments on our work!

Reviewer #2:

Comment: The authors have satisfactorily addressed all the comments. The study tested a novel hypothesis supported by solid data and a clear rationale.

Response: We thank Reviewer #2 for his/her enthusiasm and insightful comments on our work!